# Presynaptic LRP4 promotes synapse number and function of excitatory CNS neurons

Timothy J Mosca[1,2]*, David J Luginbuhl[2], Irving E Wang[3], Liqun Luo[2]

[1]Department of Neuroscience, Thomas Jefferson University, Philadelphia, United States; [2]Department of Biology, Howard Hughes Medical Institute, Stanford University, Stanford, United States; [3]Department of Neurobiology, Stanford University, Stanford, United States

**Abstract** Precise coordination of synaptic connections ensures proper information flow within circuits. The activity of presynaptic organizing molecules signaling to downstream pathways is essential for such coordination, though such entities remain incompletely known. We show that LRP4, a conserved transmembrane protein known for its postsynaptic roles, functions presynaptically as an organizing molecule. In the *Drosophila* brain, LRP4 localizes to the nerve terminals at or near active zones. Loss of presynaptic LRP4 reduces excitatory (not inhibitory) synapse number, impairs active zone architecture, and abolishes olfactory attraction - the latter of which can be suppressed by reducing presynaptic $GABA_B$ receptors. LRP4 overexpression increases synapse number in excitatory and inhibitory neurons, suggesting an instructive role and a common downstream synapse addition pathway. Mechanistically, LRP4 functions via the conserved kinase SRPK79D to ensure normal synapse number and behavior. This highlights a presynaptic function for LRP4, enabling deeper understanding of how synapse organization is coordinated.

*For correspondence: timothy.mosca@jefferson.edu

## Introduction

Multiple levels of synaptic organization ensure accurate, controlled information flow through neuronal circuits. Neurons must first make an appropriate number of synaptic connections with their postsynaptic partners. Each of these synaptic connections must have appropriate strength that can be modified by plasticity and homeostasis as a result of experience and activity changes. Further, there must be an appropriate balance between excitatory and inhibitory synapses. Finally, recent work has shown that these connections also occupy precise locations with regards to the three-dimensional structure of the synaptic neuropil. Indeed, circuit models for diverse neuronal ensembles fail to recapitulate functional patterns unless these aspects are accounted for (*Kim et al., 2014*; *Vlasits et al., 2016*). The misregulation of any one of these organizational parameters can result in neurodevelopmental disorders and intellectual disabilities like autism (*Mullins et al., 2016*), epilepsy (*Bonansco and Fuenzalida, 2016*), and other synaptopathies (*Grant, 2012*). Revealing the molecular mechanisms that ensure all of these facets are achieved is a critical step in understanding circuit assembly and function.

Synaptic organizers like Neurexins / Neuroligins, Teneurins, protein tyrosine phosphatases (PTPs), leucine rich repeat transmembrane proteins (LRRTMs), and Ephrin / Eph receptors, among others, ensure the proper number, distribution, and function of synaptic connections (*Hruska and Dalva, 2012*; *Mosca, 2015*; *Siddiqui and Craig, 2011*; *Südhof, 2008*; *Takahashi and Craig, 2013*; *de Wit and Ghosh, 2016*). Loss-of-function mutations in these key synaptogenic molecules have deleterious structural, functional, and organizational consequences for synapses and circuits. At the vertebrate

**eLife digest** The connections between nerve cells, called synapses, often malfunction in disease, injury and during aging, and to understand how this happens we first need to know how they work normally. At a synapse, one nerve cell sends a signal to the other. The signal is a chemical substance, which binds to specialized proteins called receptors on the receiving nerve cell. At excitatory synapses, the chemical signal activates the receiver; at inhibitory synapses, it does the opposite. Communication at synapses typically only goes in one direction because the sender and receiver at a synapse are not interchangeable; they contain different molecules that support their distinct roles.

To complicate matters, the same molecule may sometimes be present on both sides of a synapse with a different role in each. Moreover, not all synapses exist between two nerve cells; some synapses also form between nerve cells and muscle fibers to control the movement of the muscles. Mosca et al. set out to identify new players involved in forming synapses, and to identify differences in the formation of nerve cell-to-nerve cell versus nerve cell-to-muscle connections.

Mosca et al. were interested in particular in a protein called LRP4. In mammals, LRP4 is largely present on the muscle side of nerve cell-to-muscle synapses, where it acts as a receptor for a chemical signal called Agrin. However, fruit flies — which lack Agrin – also possess the gene for LRP4, suggesting that it has other roles too. Mosca et al. now show that LRP4 is present in the nerve cell-to-nerve cell synapses found in the fruit fly's brain. Further experiments reveal that fruit fly LRP4 plays an important role on the sender side of these synapses. Reducing the amount of LRP4 in the fruit fly brain reduces the number of excitatory, but not inhibitory, synapses. This suggests that fruit fly LRP4 may help regulate the formation of excitatory synapses.

Understanding how synapses form, and the differences between excitatory and inhibitory connections, could provide new insights into disorders of impaired synapse formation such as schizophrenia. LRP4 has also been implicated in disorders, such as amyotrophic lateral sclerosis (ALS) and myasthenia gravis, in which impaired communication between nerves and muscles causes muscles to weaken. Improved understanding of how synapses work may lead to better drugs to treat these disorders.

neuromuscular junction, one of these critical organizers is LRP4. There, it forms a receptor complex with MuSK in muscle fibers to promote clustering of acetylcholine receptors in response to motoneuron-derived agrin (*Zhang et al., 2008*; *Kim et al., 2008*; *Weatherbee et al., 2006*). Muscle LRP4 can also function as a retrograde signal with an unknown motoneuron receptor to regulate presynaptic differentiation (*Yumoto et al., 2012*). In these roles, the known functions from LRP4 are overwhelmingly postsynaptic. However, a number of lines of evidence suggest a broader role, beyond postsynaptic, for LRP4. First, motoneuron-derived LRP4 can regulate presynaptic differentiation, demonstrating a role for neuronal LRP4 (*Wu et al., 2012*). Second, in the vertebrate central nervous system (CNS), agrin is not essential for synapse formation (*Daniels, 2012*) though LRP4 can regulate synaptic plasticity, development, and cognitive function (*Gomez et al., 2014*; *Pohlkamp et al., 2015*), through functioning in astrocytes in some cases (*Sun et al., 2016*). In this vein, the *Drosophila* genome contains an LRP4 homologue, but no clear agrin or MuSK homologues (*Adams et al., 2000*), so any role for LRP4 there must be agrin-independent.

Here, we show in the *Drosophila* CNS that LRP4 is a presynaptic protein that regulates the number, architecture, and function of synapses. LRP4 functions largely through the conserved, presynaptic SR-protein kinase, SRPK79D. LRP4 and SRPK79D interact genetically and epistatically, as SRPK79D overexpression can suppress *lrp4*-related phenotypes. Unexpectedly, this role for LRP4 occurs preferentially in excitatory neurons, as impairing *lrp4* in inhibitory neurons has no effect. As little is known about the presynaptic determinants (save neurotransmitter-related enzymes and transporters) of excitatory versus inhibitory synapses, this may suggest a new mode for distinguishing such synapses from the presynaptic side. Thus, LRP4 may represent a conserved synaptic organizer that functions presynaptically, cell autonomously, and independently of agrin to coordinate synapse number and function.

## Results

### LRP4 is a synaptic protein expressed in excitatory neurons

We identified CG8909 as the fly LRP4 homologue (*Figure 1—figure supplements 1* and *2A*), which is predicted to be a single-pass transmembrane protein whose domain organization resembles that of mammalian LRP4 (*Figure 1A*). *Drosophila* LRP4 shares 38% identity with human LRP4 overall, 61% identity within the LDL-repeat containing extracellular portion, and 28% identity in the intracellular tail. Consistent with previous expression data from whole-brain microarrays (*Chintapalli et al., 2007*), we determined that LRP4 was expressed throughout the adult brain using antibodies against the endogenous protein (*Figure 1B–C*) or an *lrp4-GAL4* transgene that expresses GAL4 under the *lrp4* promoter and visualized with either Syt-HA (*Figure 1D*) or an HA epitope-tagged LRP4 (*Figure 1—figure supplement 2C*). All methods revealed similar patterns of expression in the antennal lobes (*Figure 1* and *Figure 1—figure supplement 2C–E*), optic lobes, and higher olfactory centers including the mushroom body and the lateral horn (*Figure 1B,D*). Antibody specificity was validated by the complete loss of signal in a deletion (see below) of the *lrp4* coding region (*Figure 1C*). We further investigated LRP4 in the antennal lobe, the first olfactory processing center in the *Drosophila* CNS, which has emerged as a model circuit for studying sensory processing (*Wilson, 2013*) and whose synaptic organization was recently mapped at high resolution (*Mosca and Luo, 2014*).

LRP4 was enriched in the synaptic neuropil of the antennal lobe (*Figure 1B*). As this neuropil is made up of processes from multiple classes of olfactory neurons, all of which make presynaptic connections there, we used intersectional strategies with *lrp4-GAL4* to identify which neurons expressed *lrp4*. These approaches revealed *lrp4* expression in both olfactory receptor neurons (ORNs; *Figure 1—figure supplement 2D*) and projection neurons (PNs; *Figure 1—figure supplement 2E*). Because of the observed neuropil expression of LRP4 (*Figure 1B–C*), we sought to examine the localization of LRP4 with regards to a known synaptic protein, the active zone scaffolding component Bruchpilot (*Wagh et al., 2006*). However, due to the density of CNS neuropil, colocalization analyses using light level microscopy have inherently low resolution. Therefore, we applied expansion microscopy (*Chen et al., 2015*) to the *Drosophila* CNS to improve the resolution of colocalization analysis. This technique uses isotropic expansion of immunolabeled tissue (*Tillberg et al., 2016*) while maintaining the spatial relationship between protein targets and allowing for enhanced resolution with confocal microscopy. Using protein-retention expansion microscopy (proExM), we obtained reliable, ~4 fold isotropic expansion of *Drosophila* CNS tissue (*Figure 1—figure supplement 3*). To specifically examine the relationship between LRP4 and active zones only in ORNs, we expressed HA-tagged LRP4 and Brp-Short-mStraw using the *pebbled-GAL4* driver (*Sweeney et al., 2007*). LRP4-HA expressed using *lrp4-GAL4* localizes to similar regions as LRP4 antibody staining (*Figure 1B* and *Figure 1—figure supplement 2C*), suggesting the fidelity of this transgene. Within individual expanded glomeruli of proExM-treated brains, LRP4 and Brp localized to similar regions (*Figure 1E*) and, when examined at high magnification, LRP4 localized either coincidentally with Brp (*Figure 1F*, arrowhead) or to the space adjacent to active zones (*Figure 1F*, arrow). This combination of active zone and periactive zone localization is similar to that of known synaptic organizers (*Jepson et al., 2014*; *Li et al., 2007*; *Mosca et al., 2012*). Thus, LRP4 is a synaptic protein that localizes to nerve terminals.

Given widespread expression throughout the brain, we sought to identify the cell types that express LRP4. To accomplish this, we used *lrp4-GAL4* driven mCD8-GFP as this approach, in addition to labeling similar neuropil regions as the antibody, also highlighted the cell bodies of *lrp4*-positive cells. We co-stained brains for various cellular and neuronal-subtype markers and quantified the overlap between cells positive for *lrp4*-expression and expression of these various labels. Nearly all *lrp4*-positive cells observed (99.5%) expressed the neuronal marker ELAV (*Robinow and White, 1988*) (*Figure 1G*), indicating that these cells were neurons. Few (0.4%) expressed the glial marker Repo (*Xiong et al., 1994*) (*Figure 1H*). The majority of *lrp4*-positive cells (59.1%) also expressed choline acetyltransferase (ChAT; *Figure 1I*), a marker for cholinergic excitatory neurons. We also observed partial overlap between *lrp4*-positive neurons and vGlut (22.4%; *Figure 1J*), the vesicular transporter for glutamate. In the fly brain, glutamatergic neurons can be either excitatory or inhibitory (*Liu and Wilson, 2013*). Interestingly, there was little overlap (0.3%) between *lrp4* and GABA, the major inhibitory neurotransmitter in *Drosophila* (*Figure 1K*). Thus, LRP4 is expressed at synaptic

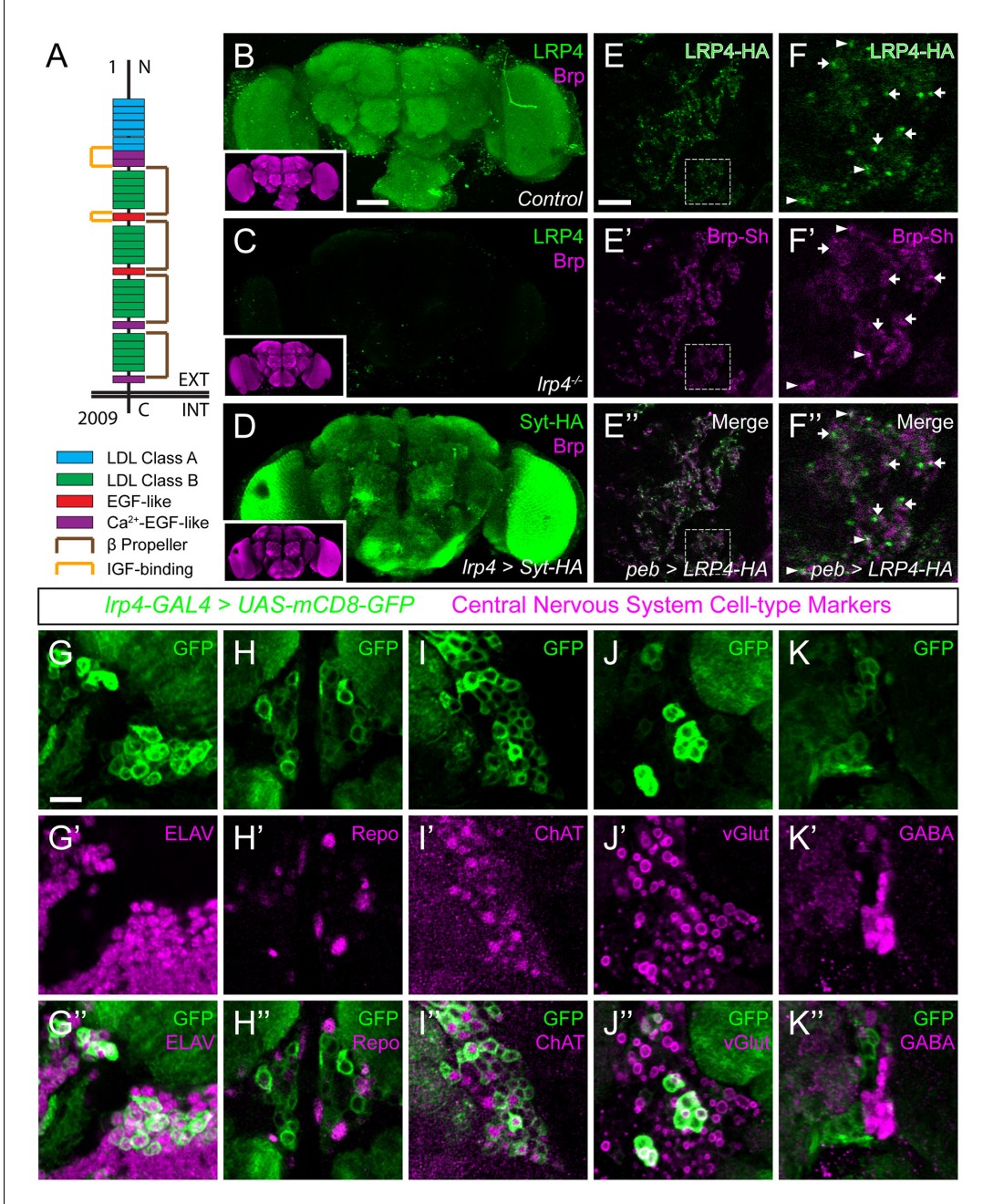

**Figure 1.** LRP4 is a synaptic protein expressed in excitatory neurons. (A) Domain structure of *Drosophila* LRP4. Numbers indicate amino acids. EXT, extracellular side. INT, intracellular side. (B) Representative confocal image stack of a control *Drosophila* brain stained with antibodies against endogenous LRP4 (green) and Bruchpilot (inset, magenta) demonstrating expression throughout the brain. (C) Representative confocal image stack of an *Irp4^dalek* null brain stained with antibodies against LRP4 (green) and Brp (inset, magenta) demonstrating antibody specificity. (D) Representative confocal image of a *Drosophila* brain expressing *UAS-Syt-HA* via *lrp4-GAL4* and stained with antibodies to HA (D, green) and N-Cadherin (inset, magenta). The expression pattern resembles that of endogenous LRP4, supporting the specificity of *lrp4-GAL4.* (E) Representative single slice within a single antennal lobe glomerulus of a brain processed for expansion microscopy (proExM) expressing LRP4-HA and Brp-Short-mStraw in all ORNs via *pebbled-GAL4* and stained with antibodies to HA (E, E″, green) and mStraw (E′-E″, magenta). LRP4 localizes to synaptic neuropil regions. (F) High magnification image of the region bounded by dashed lines in (E) and stained as above. Arrows indicate LRP4-HA localization adjacent to / not directly overlapping with Bruchpilot-Short. Arrowheads indicate overlapping LRP4-HA and Brp-Short localization. (G–K) Representative high magnification confocal stack images of neuronal cell bodies surrounding the antennal lobe in animals expressing *UAS-mCD8-GFP* via *lrp4-GAL4* and stained for antibodies against GFP (G-K, green) and other cell-type markers (G′-K′, magenta). Merge channels (G″–K″) show colocalization of *lrp4* with the neuronal marker ELAV (G″) but not the glial cell marker Repo (H″). Neurons positive for *lrp4* show colocalization with choline acetyltransferase (ChAT,

*Figure 1 continued on next page*

*Figure 1 continued*

I''), and the vesicular glutamate transporter (vGlut, **J''**), but little to no colocalization with the inhibitory neurotransmitter GABA (**K''**), suggesting that *lrp4*-positive cells are largely excitatory neurons. The percentage of GFP-positive cells that are ALSO positive for the cell-type specific marker are as follows: Elav = 99.50 ± 0.19% overlap; Repo = 0.38 ± 0.18% overlap; ChAT = 59.13 ± 2.48% overlap; vGlut = 22.38 ± 1.28% overlap; GABA = 0.25 ± 0.16% overlap. For all cases, *n* = 8 animals, ≥ 200 cells per animal. Values = mean ± s.e.m. Scale bars = 50 µm (**B–D**), 150 µm (B-D, insets), 25 µm (**E–F**), 10 µm (**G–K**).

The following figure supplements are available for figure 1:

**Figure supplement 1.** Sequence alignment of *Drosophila*, mouse, and human LRP4 homologues.

**Figure supplement 2.** LRP4 reagents and patterns of LRP4 expression.

**Figure supplement 3.** Validation of expansion microscopy in *Drosophila*.

terminals of a subset of excitatory cholinergic neurons and a subset of glutamatergic neurons that may be excitatory or inhibitory, but is excluded from inhibitory GABAergic neurons.

## Perturbing presynaptic LRP4 changes ORN synapse number

As both the expression and localization of LRP4 were consistent with the protein serving a synaptic role, we sought to determine whether disrupting its function in excitatory neurons would affect synapse number. To image these connections, we expressed fluorescently tagged synaptic markers (*Fouquet et al., 2009*; *Leiss et al., 2009*; *Mosca and Luo, 2014*) and used previously established methods to estimate the number of active zones and postsynaptic receptor puncta (*Mosca and Luo, 2014*) in olfactory neurons in antennal lobe glomeruli (*Figure 2A*). These methods show stereotyped active zone numbers and densities in ORNs and can reveal the function of synaptic proteins in mediating these aspects (*Mosca and Luo, 2014*). Further, measurements from these methods are consistent with our own electron microscopy (*Mosca and Luo, 2014*) as well as results from ultrastructural reconstructions of all synapses in individual glomeruli (*Tobin et al., 2017*) demonstrating their utility. To perturb LRP4 function, we created a null mutation (*lrp4^dalek^*) using the CRISPR-Cas9 system (*Gratz et al., 2013*) that removed the entire coding region (*Figure 1—figure supplement 2A–B*). *lrp4^dalek^* mutants were viable with a slightly reduced body size.

In ORN axon terminals projecting to the VA1v glomerulus in males (*Figure 2B*), *lrp4^dalek^* mutants (*Figure 2C,H*) showed a 31% reduction in the number of puncta for Brp-Short, an active zone marker, compared to control adults (*Figure 2B,H*). This phenotype was recapitulated when we expressed any of four independent transgenic RNAi constructs against *lrp4* only in ORNs (*Figure 2D,H*, and *Figure 2—figure supplement 1*), demonstrating that LRP4 functions presynaptically in regulating active zone number. These changes were independent of glomerular volume: *lrp4* loss-of-function had no effect on neurite volume (*Figure 2H* and *Figure 2—figure supplement 1*). Though the intensity of Brp-Short puncta across some genotypes trended slightly downward, it did not reach statistical significance (data not shown). We also observed that *lrp4* disruption (using *lrp4^dalek^* mutants and presynaptic RNAi expression) caused a quantitatively similar reduction of active zone numbers in VA1v ORN axon terminals in females in this sexually dimorphic glomerulus (*Figure 2—figure supplement 2*), and in ORN axon terminals projecting to the VA1d, DA1, DL4, and DM6 glomeruli (*Figure 2—figure supplement 3*). This suggests that *lrp4* phenotypes are not specific to particular glomeruli. Beyond Brp-Short, we observed similar phenotypes with an independent presynaptic marker, DSyd-1 (*Owald et al., 2012*), that is also punctate at ORN terminals (*Mosca and Luo, 2014*) (*Figure 2—figure supplement 4*).

We further examined the consequences of *lrp4* disruption on the number of Dα7 acetylcholine receptor puncta in PN dendrites postsynaptic to the ORN axon terminals imaged above. Loss of *lrp4* decreased Dα7-EGFP puncta numbers by 29% compared to controls (*Figure 2F–G,I*). This deficit was also independent of neurite volume (*Figure 2I* and *Figure 2—figure supplement 2*), again demonstrating that *lrp4* perturbation phenotypes did not result from decreased neuronal projection size. Further, both the presynaptic active zone and postsynaptic acetylcholine receptor phenotypes were quantitatively similar. While likely that the postsynaptic AChR number decreases concomitantly with

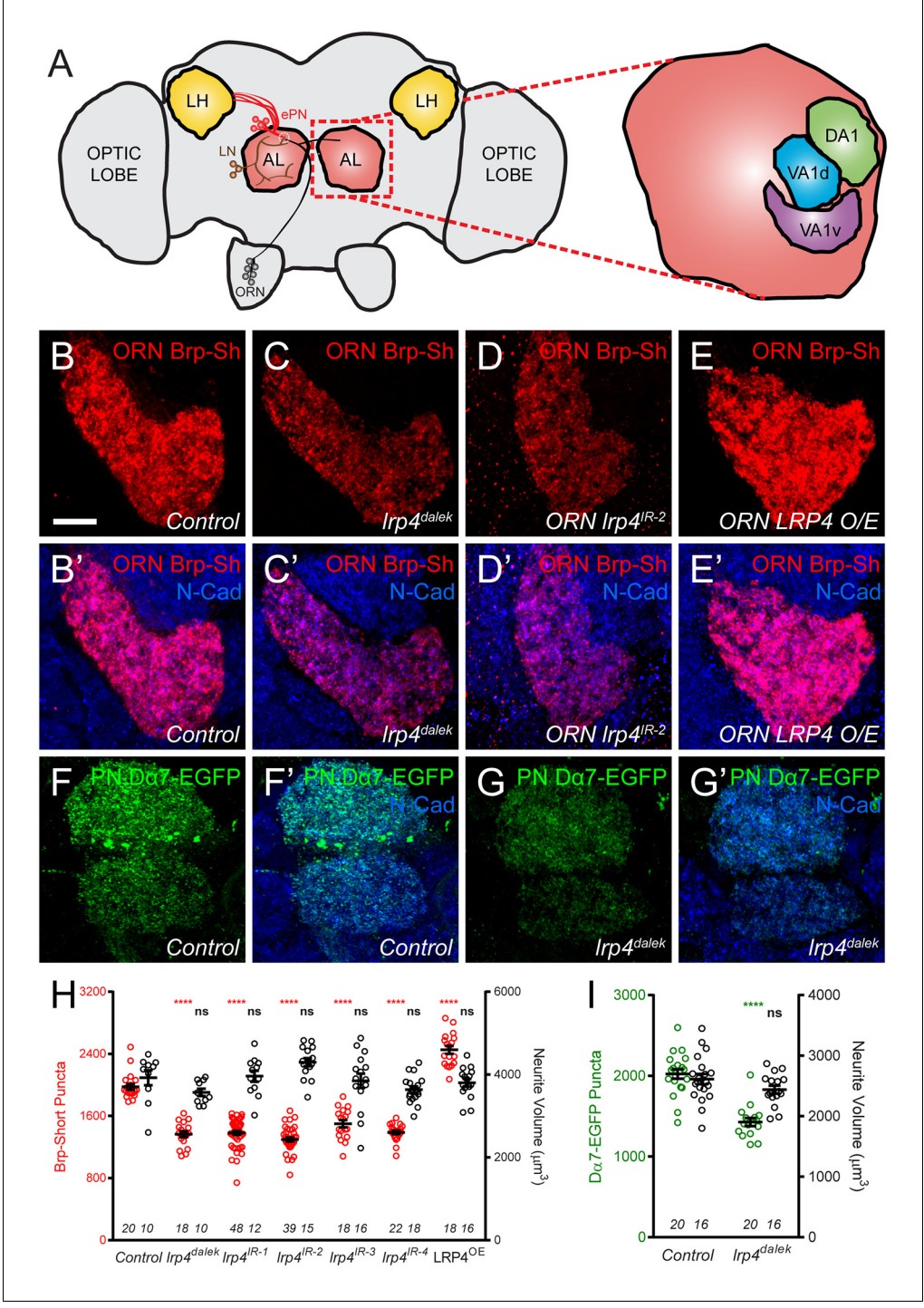

**Figure 2.** LRP4 perturbation in excitatory neurons alters synapse number. (**A**) Schematic diagram of the fly brain with major regions labeled and the olfactory regions examined in this study shaded in red (AL, antennal lobe) or yellow (LH, the lateral horn). Olfactory receptor neurons (ORNs, black), excitatory projection neurons (ePNs, red), and local interneurons (LNs, brown) are indicated. White dashed lines represent a glomerulus. Magnification: the antennal lobe region with the three glomeruli examined here highlighted: DA1 (green), VA1d (blue), and VA1v (purple). (**B–E**) Representative high magnification confocal stack images of VA1v ORN axon terminals in the VA1v glomerulus of males expressing Brp-Short-mStraw and stained with antibodies against mStraw (red) and N-Cadherin (blue). Loss of *lrp4* (*lrp4^dalek^*) and RNAi against *lrp4* expressed only in ORNs (*ORN lrp4^IR-2^*) show fewer Brp-Short-mStraw puncta while LRP4 overexpression in ORNs (*ORN LRP4 OE*) increases the number of Brp-Short-mStraw puncta. (**F–G**) Representative high magnification confocal maximum intensity projections of DA1 and VA1d

*Figure 2 continued*

PN dendrites in males expressing Dα7-EGFP, a tagged acetylcholine receptor subunit. Loss of *lrp4 (lrp4^dalek)* also results in fewer Dα7-EGFP puncta. (H) Quantification of Brp-Short-mStraw puncta (red, left axis) and neurite volume (black, right axis) in VA1v ORNs. (I) Quantification of Dα7-EGFP puncta (green, left axis) and neurite volume (black, right axis). ****p<0.0001; ***p<0.001; ns, not significant. Statistical comparisons in 2H (one-way ANOVA with correction for multiple comparisons) are with control. Statistical comparisons between two samples are done via Student's t-test. Error bars represent mean ± s.e.m. *n* (antennal lobes) is noted at the bottom of each column. Scale bars = 10 μm.

The following figure supplements are available for figure 2:

**Figure supplement 1.** Representative antennal lobe images for genetic *lrp4* manipulations.

**Figure supplement 2.** *lrp4* perturbation in females affects synapse number.

**Figure supplement 3.** *lrp4* RNAi reduces synapse number in multiple glomeruli.

**Figure supplement 4.** *lrp4* RNAi reduces Syd1 puncta in presynaptic ORN terminals.

---

the presynaptic active zone number (which is controlled by presynaptic LRP4), we cannot exclude an additional postsynaptic role for LRP4 (see Discussion). However, it is evident that the loss of LRP4 reduces synapse number as assayed both pre- and postsynaptically.

The above experiments demonstrated the necessity of presynaptic LRP4 in ensuring the proper number of synaptic connections. However, with known presynaptic organizers like Neurexin, overexpression results in added boutons (*Li et al., 2007*) and active zones (*Craig and Kang, 2007*). To test for LRP4 sufficiency in synapse addition, we overexpressed HA-tagged LRP4 presynaptically in otherwise wild-type ORNs. LRP4 overexpression increased the number of Brp-Short puncta by 30% (*Figure 2E,H*, and *Figure 2—figure supplement 2*); this increase was also independent of neurite volume (*Figure 2h* and *Figure 2—figure supplements 2–3*) as the glomeruli remained the same size. Thus, there is a direct relationship between presynaptic LRP4 expression and synapse number in excitatory neurons: removing LRP4 reduces, while overexpressing LRP4 increases, synapse number.

## Ultrastructural analysis reveals LRP4 regulates active zone number and structure

Though light level analyses accurately report fold-changes in synapse number (*Chen et al., 2014*; *Mosca and Luo, 2014*), we sought to independently confirm and extend our analyses using electron microscopy. Using transmission electron microscopy (TEM) on the fly antennal lobe, we quantified synapse number in putative ORN terminals based on morphology (*Rybak et al., 2016*; *Tobin et al., 2017*) in both control (*Figure 3A*) and *lrp4^dalek* (*Figure 3B*) adult brains. T-bar profiles were evident in both genotypes, but were reduced in number by 31% in mutant terminals (*Figure 3C*), which exactly matched the reduction observed by Brp-Short puncta measurements (*Figure 2H*). Terminal perimeter was slightly but significantly increased in *lrp4^dalek* terminals (*Figure 3D*), resulting in a 36% reduction in T-bar density when compared to control (*Figure 3E*). These results are consistent with those observed via confocal microscopy, and demonstrate that LRP4 is necessary for the proper number of synapses in putative ORN terminals of the antennal lobe.

Brp-Short assays alone cannot distinguish between normal and impaired active zones. We therefore examined the ultrastructural morphology of individual active zones to determine if LRP4 had an additional role in the biogenesis of the T-bar itself. In both control (*Figure 3F–H*) and *lrp4* (*Figure 3I–K*) terminals, we observed single (*Figure 3F–I*), double (*Figure 3G,J*), and triple T-bars (*Figure 3H,K*) suggesting that LRP4 is not absolutely required for T-bar formation and some elements of organization. However, whereas irregular T-bars in control animals were rare (<5% of total T-bars), the majority of T-bars in *lrp4* mutants displayed one or more defects (*Figure 3L–Q*), including immature T-bars that lacked tops (*Figure 3L*), detached T-bars (*Figure 3M*), misshapen T-bars of varying configurations and aggregations (*Figure 3N–P*), and multiple T-bars beyond those observed

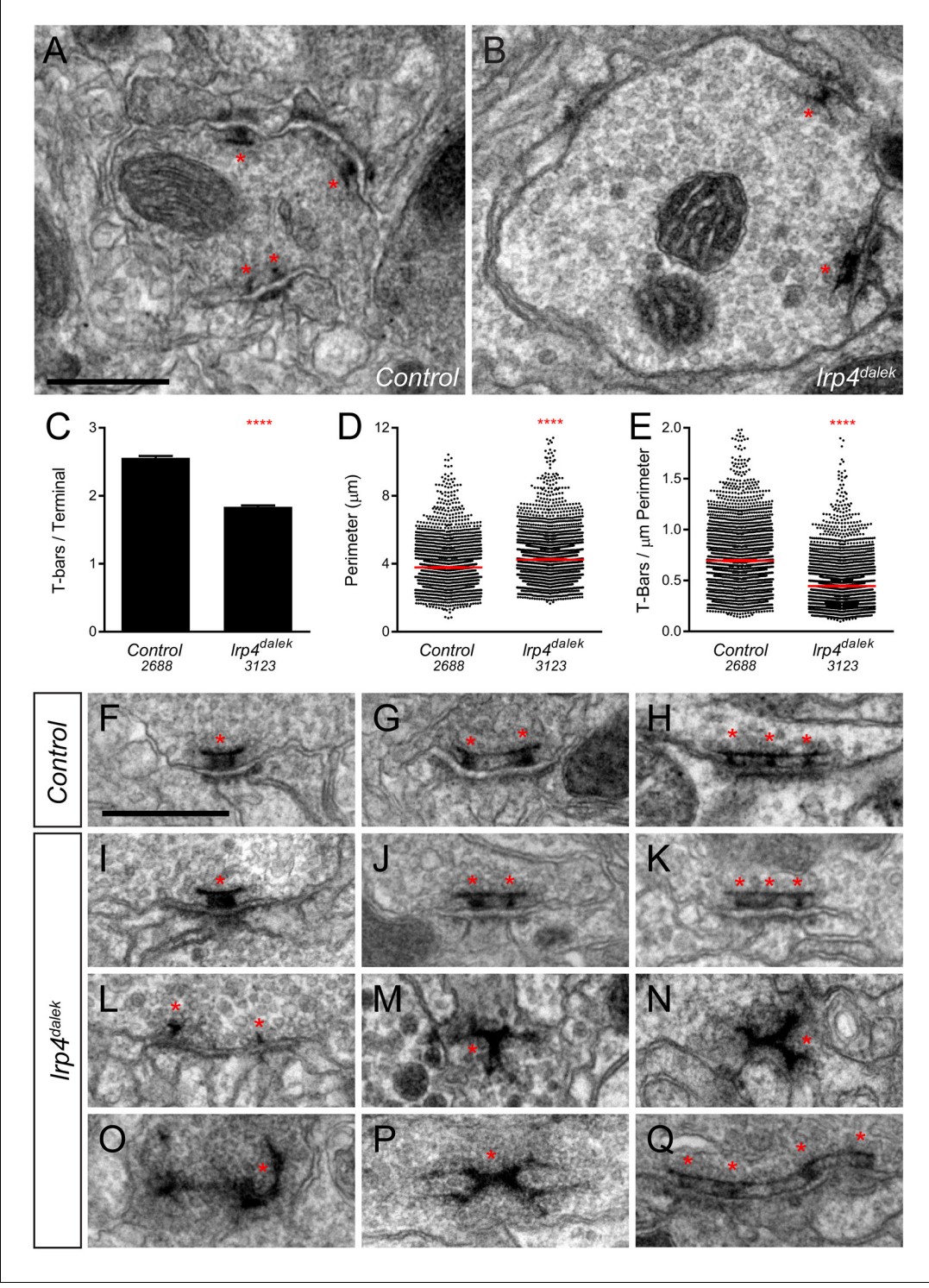

**Figure 3.** Loss of LRP4 causes defects in T-bar number and morphology. (**A–B**) Representative transmission electron micrographs of putative ORN terminal in *Control* (**A**) and *Irp4*<sup>*dalek*</sup> (**B**) adult antennal lobes. Loss of *Irp4* results in fewer observed T-bar profiles (asterisk) and a larger terminal perimeter. Scale bar = 1 µm. (**C**) Quantification of T-bar profiles per terminal in *Control* and *Irp4*<sup>*dalek*</sup> terminals. Loss of LRP4 results in a 31% reduction of T-bars. (**D**) Quantification of terminal perimeter in *Control* and *Irp4*<sup>*dalek*</sup> adults. Mutant terminals have a 13% greater perimeter than control terminals. (**E**) Quantification of the T-bar density per µm of terminal perimeter. Loss of LRP4 causes a 36% reduction in T-bar density when the increased terminal perimeter is accounted for. For (**C–E**), Control has *n* = 5 animals, 2688 terminals and *Irp4*<sup>*dalek*</sup> has *n* = 3 animals, 3123 terminals. *Figure 3 continued on next page*

*Figure 3 continued*

The number of terminals measured is listed below the genotype. ****p<0.0001. Statistical comparisons (two-tailed Student's t-test) are done between genotypes. Error bars represent mean ± s.e.m. (F–H) Representative transmission electron micrographs of individual T-bar profiles (asterisk) in *control* adults. Single (F), double (G), and triple (H) profiles are readily visible. (I–Q) Representative transmission electron micrographs of individual T-bar profiles in *lrp4^dalek^* adults. As in control flies, single (I), double (J) and triple (K) T-bar profiles were visible. The majority of T-bars, however, demonstrated morphology defects including those that lacked table tops (L), were detached from the membrane (M–N), were misshapen (N–P), and profiles containing four or more connected T-bars (Q). These all represent morphological defects that are not observed (or very rarely observed) in control adults. Scale bar = 200 nm.

in control animals (*Figure 3Q*). Thus, in addition to controlling the number of synapses, LRP4 is also required for individual active zones to assume normal morphology, attach to the membrane, and have proper spacing. Thus, LRP4 has multiple, critical roles in central synapse formation.

## LRP4 is not required for inhibitory neuron synapse number

The preferential expression of *lrp4* in excitatory but not inhibitory neurons (*Figure 1*) suggests that it promotes synapse addition specifically in excitatory neurons. To test this, we used Brp-Short to examine synapse number in GABAergic inhibitory neurons projecting to the antennal lobe using the *GAD1-GAL4* driver (*Ng et al., 2002*). Though GAD1-positive neurons project throughout the antennal lobe (*Mehren and Griffith, 2006*), we restricted our analyses to the DA1 glomerulus, where we observed reductions in excitatory synapses (*Figure 2—figure supplements 3–4*) following LRP4 disruption. When LRP4 function was impaired using the *lrp4^dalek^* mutant or RNAi in these neurons, synapse number was unaffected (*Figure 4A–B,D*). Thus, the reduction of synapse number under LRP4 loss-of-function conditions appeared specific for excitatory neurons.

Interestingly, when LRP4 was overexpressed in inhibitory neurons, we observed a 35% increase in synapse number without an accompanying change in neurite volume, similar to what we observed for excitatory neurons (*Figure 4C–D*). This suggests that, while inhibitory GABAergic neurons do not normally utilize LRP4 to regulate synapse number, they possess the downstream machinery necessary for LRP4 to function in adding synapses. Thus, when LRP4 is exogenously expressed in these cells, it can co-opt this machinery for synapse addition. As such, excitatory and inhibitory neurons likely use distinct cell surface synaptic organizers (LRP4 for excitatory neurons) that converge on common mechanisms for synapse addition.

## Excitatory, but not inhibitory, olfactory projection neurons also require LRP4 to ensure proper synapse number

Though we initially restricted our analyses to the antennal lobe, we also observed *lrp4* expression throughout the brain, including two higher order olfactory neuropil: the mushroom body and the lateral horn (*Figure 1B–D*). To determine whether LRP4 could also serve as a synaptic organizer in these brain regions, we examined the effects of *lrp4* perturbation on both excitatory and inhibitory synapses in the lateral horn (LH, *Figure 5A*), a higher order olfactory center involved in innate olfactory behavior (*Heimbeck et al., 2001*). We used *Mz19-GAL4* to label projection neurons whose dendrites and cell bodies are restricted to the antennal lobe region, but whose axon terminals make excitatory synapses in the lateral horn (*Berdnik et al., 2006*). To label inhibitory synapses, we used the *Mz699-GAL4* driver, which is expressed in inhibitory projection neurons (iPNs) whose dendrites project to the antennal lobe and whose axons project to the lateral horn (*Lai et al., 2008*; *Liang et al., 2013*). *Mz699-GAL4* also labels a small subset of third-order neurons that project dendrites largely void of presynaptic terminals to the ventral lateral horn (*Liang et al., 2013*). Thus, we consider synaptic signal labeled by *Mz699-GAL4* as being contributed mostly by iPNs.

In *lrp4* mutants, the number of excitatory lateral horn synapses was reduced by 40%, consistent with a role for LRP4 in synapse formation (*Figure 5B–C,F*). PN perturbation of *lrp4* using RNAi reduced synapse number similarly to the loss-of-function allele, demonstrating a presynaptic role for *lrp4* in these neurons (*Figure 5F* and *Figure 5—figure supplement 1*). These changes were independent of neurite volume, which remained unaffected (*Figure 5F*). Perturbation of *lrp4* in *Mz699-*

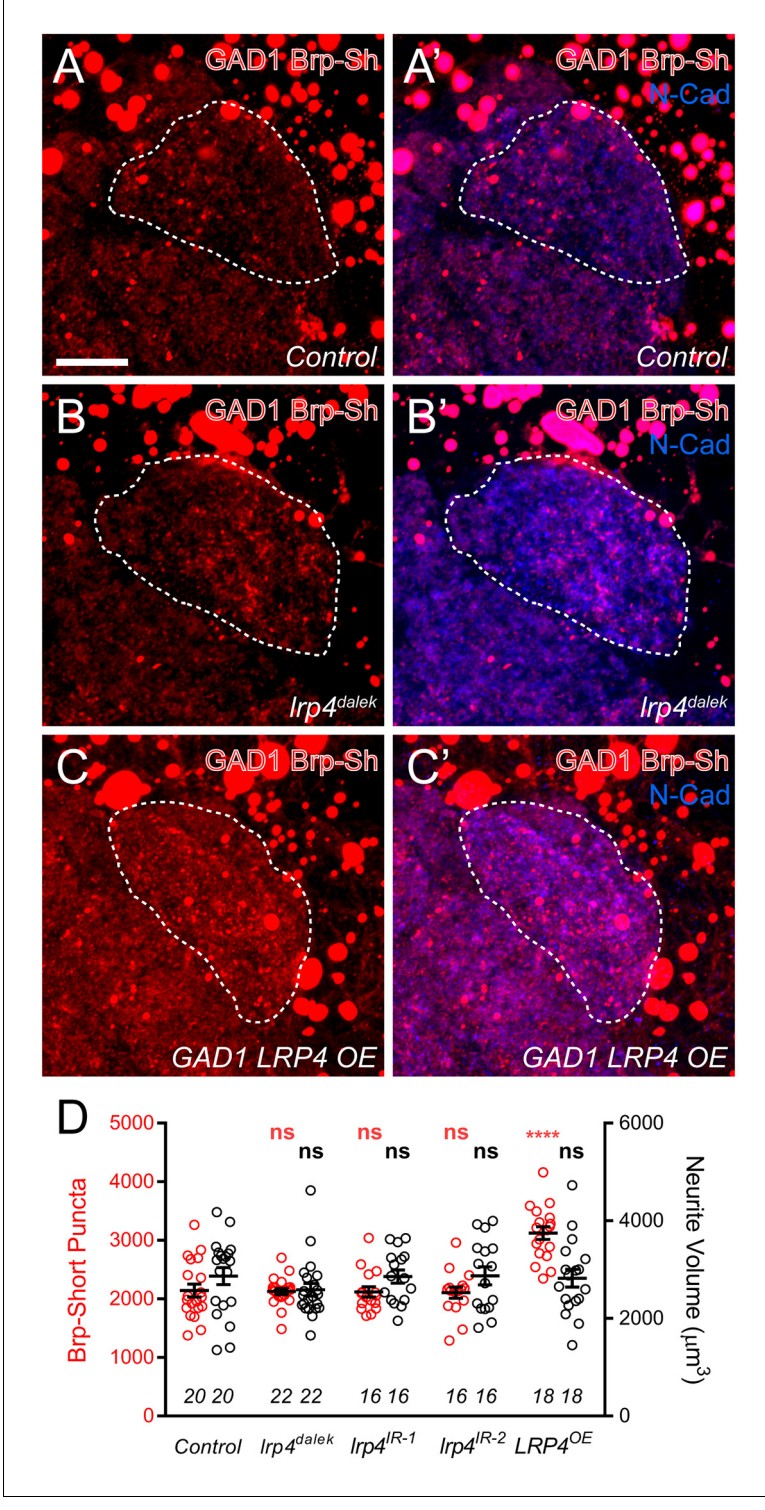

**Figure 4.** Effects of LRP4 perturbation on inhibitory neuron synapse formation. (**A–C**) Representative high magnification confocal maximum intensity projections of GAD1-positive inhibitory neurons, which project to the DA1 glomerulus (dashed line), in males expressing Brp-Short-mStraw and stained with antibodies against mStraw (red) and N-Cadherin (blue). Due to the proximity of inhibitory neuron cell bodies to the antennal lobe, saturated somatic signal is observed. Loss of *lrp4* (*lrp4^dalek*) does not affect puncta number but overexpression of LRP4 (*GAD1 LRP4 OE*) increases Brp-Short puncta. (**D**) Quantification of Brp-Short-mStraw puncta (red, left axis) and neurite volume (black, right axis) in GAD1 neurons. Neither loss of *lrp4* nor RNAi against *lrp4* expressed in inhibitory neurons affects puncta number or neurite volume. ****p<0.0001; ***p<0.001; ns, not significant.
*Figure 4 continued on next page*

*Figure 4 continued*

Statistical comparisons (one-way ANOVA with correction for multiple comparisons) are with control. Error bars represent mean ± s.e.m. *n* (antennal lobes) is noted at the bottom of each column. Scale bars = 10 μm.

positive iPNs, however, had no effect on the number of synapses (*Figure 5D–E,G*, and *Figure 5— figure supplement 1*) despite a slight reduction in neurite volume in *lrp4*$^{dalek}$ mutants (*Figure 5G*). Despite a lack of a loss-of-function phenotype, we observed an increase in synapse number when we overexpressed LRP4-HA in *Mz699*-positive neurons (*Figure 5G* and *Figure 5—figure supplement 1*). Thus, the results of *lrp4* perturbation on excitatory and inhibitory synapses in the lateral horn resembled those of the antennal lobe, suggesting a general role for LRP4 in promoting excitatory synapse number.

## LRP4 is required for normal olfactory attraction behavior

Given the role for LRP4 in the specific regulation of excitatory synapse number, we sought to determine whether the consequences of LRP4 disruption were accompanied by functional changes in behavior. We examined fly attraction to the odorant in apple cider vinegar using a modified olfactory trap assay (*Larsson et al., 2004*; *Potter et al., 2010*) (*Figure 6A*), an ethologically relevant assay that requires flight and/or climbing to follow odorant information within a larger arena (*Min et al., 2013*). As presynaptic LRP4 regulates ORN synapse number, we used RNAi against *lrp4* expressed selectively in all ORNs using *pebbled-GAL4* to assess olfactory attraction. Control flies bearing a single copy of *pebbled-GAL4* or one of four different *lrp4* RNAi transgenes alone exhibited a strong preference for apple cider vinegar (*Figure 6B*). Flies bearing both transgenes (and thus, reduced *lrp4* expression) exhibited a near complete abrogation of attractive behavior and were no longer able to distinguish the attractive apple cider vinegar from a water control (*Figure 6B*). Movement, wall climbing, and flight were still observed in these flies (data not shown), suggesting that this was not due to widespread defects in motion, consistent with our selective perturbation of LRP4 function in ORNs. Thus, presynaptic LRP4 in ORNs is necessary for normal olfactory attraction behavior.

A complete loss of olfactory attraction was unexpected for a manipulation that reduced synapse number by ~30%. One potential explanation is that, while the remaining 70% of synapses were detected by the Brp-Short assay, they were functionally impaired. This would be consistent with the myriad of morphology defects observed in *lrp4* mutant T-bars via TEM (*Figure 3I–Q*). In *Drosophila*, olfactory information flow is regulated by presynaptic inhibition by local GABAergic interneurons onto excitatory ORNs via the GABA$_A$ and GABA$_B$R2 receptors (*Olsen and Wilson, 2008*; *Root et al., 2008*). If the remaining synapses were indeed weakened by the loss of LRP4, reducing inhibition onto those ORNs might suppress the behavioral phenotype. To test this hypothesis, we inhibited the GABA$_B$R2 receptor in ORNs using RNAi, which by itself did not affect the olfactory attraction behavior (*Figure 6B*). Simultaneous knockdown of *GABA$_B$R2* and *lrp4*, however, markedly suppressed the behavioral phenotype associated with *lrp4* knockdown alone (*Figure 6B*). This manipulation did not suppress the morphological phenotype, however, as the reduction in Brp-Short puncta was still apparent (1297 ± 25.62 puncta, *n* = 39 antennal lobes for *Or47b-GAL4 > UAS-lrp4*$^{IR2}$ *+ UAS-mCD8-GFP* vs. 1191 ± 48.91 puncta, *n* = 12 antennal lobes for *Or47b-GAL4 > UAS-lrp4*$^{IR2}$ *+ UAS-GABABR2*$^{IR}$, p>0.2). These results suggest that olfactory attraction behavior requires a proper level of net excitatory drive in the antennal lobe circuit and that defects caused by weakened excitatory synapses can be compensated for by reducing inhibition.

## SRPK79D interacts with, and requires, LRP4 for ORN terminal localization

To understand how LRP4 could regulate excitatory synapse number and olfactory behavior, we investigated the mechanism by which it functions. In examining *lrp4*$^{dalek}$ mutant larvae and larvae where *lrp4* was specifically knocked down in all neurons using RNAi, we observed impaired localization of active zone material (*Figure 7A–C*). Under normal circumstances, the active zone marker Bruchpilot (*Wagh et al., 2006*) and the synaptic vesicle marker Synaptotagmin I (*DiAntonio et al.,*

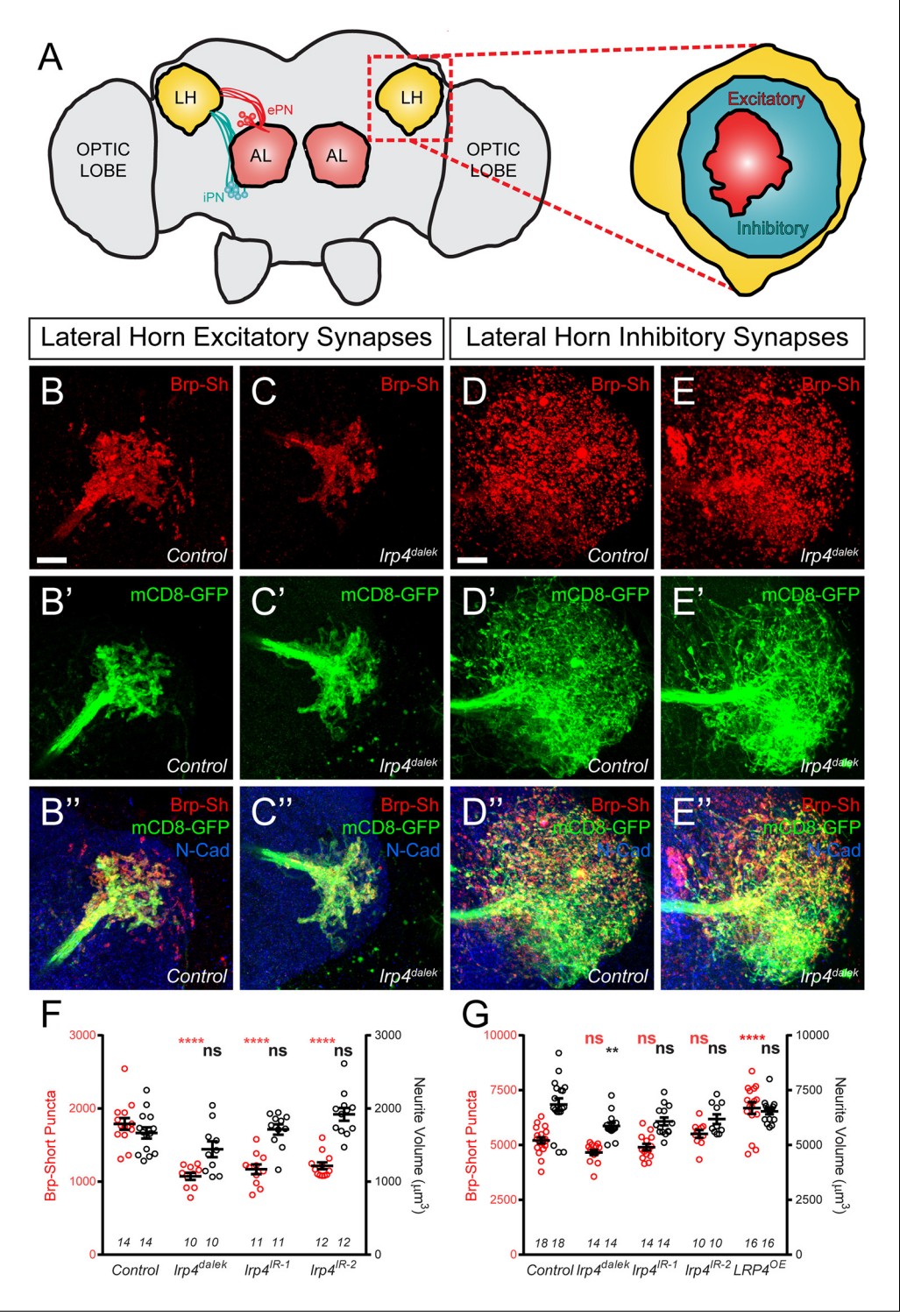

**Figure 5.** LRP4 perturbations similarly affect higher order olfactory centers. (**A**) Schematic diagram of the fly brain with major regions labeled and the olfactory regions examined in this study shaded in red (AL, antennal lobe) or yellow (LH, the lateral horn). Excitatory projection neuron (ePN, dark red) and inhibitory projection neuron (iPN, teal) axons are indicated. Magnification: the lateral horn region with the regions innervated by excitatory Mz19-positive projection neuron axons (ePNs, dark red) and inhibitory Mz699-positive projection neuron axons (iPNs, teal) examined here highlighted. (**B–C**) Representative high magnification confocal maximum intensity projections of *Mz19-GAL4* positive PN axon terminals in the lateral horn in males expressing Brp-Short-mStraw and mCD8-GFP and stained for antibodies against mStraw (red), GFP (green), and N-Cadherin (blue). Loss of *lrp4* (**B**, *lrp4^dalek^*)

*Figure 5 continued*

reduces synapse number compared to control (**A**). (**D–E**) Representative high magnification confocal maximum intensity projections of *Mz699-GAL4* positive inhibitory projection neuron (iPN) axon terminals in the lateral horn in males expressing Brp-Short-mStraw and mCD8-GFP and stained for antibodies against mStraw (red), GFP (green), and N-Cadherin (blue). Loss of *lrp4* (E, *lrp4^dalek*) does not affect synapse number compared to control (**D**). (**F**) Quantification of Brp-Short-mStraw puncta (red, left axis) and neurite volume (black, right axis) in Mz19-positive excitatory projection neurons. Loss of *lrp4* and RNAi against *lrp4* expressed in those neurons reduces puncta number but leaves neurite volume unaffected. The similar reduction in puncta number between mutants and PN-specific RNAi reveals the cell autonomous nature of the *lrp4* phenotype. (**G**) Quantification of Brp-Short-mStraw puncta (red, left axis) and neurite volume (black, right axis) in Mz699-positive inhibitory projection neurons. Neither loss of *lrp4* nor *lrp4* RNAi expressed in those neurons affects puncta number, similar to inhibitory neurons in the antennal lobe. Loss of *lrp4* reduces neurite volume by 11% but RNAi does not. Overexpression of LRP4 in these neurons (*LRP4 OE*) results in a 28% increase in the number of Brp-Short puncta. ****p<0.0001; **p<0.01; ns, not significant. Statistical comparisons (one way ANOVA with correction for multiple comparisons) are with control. Error bars represent mean ± s.e.m. *n* (lateral horns) is noted at the bottom of each column. Scale bars = 10 μm.

The following figure supplement is available for figure 5:

**Figure supplement 1.** Representative lateral horn images for LRP4 genetic manipulations.

---

*1993*) were barely detectable in larval transverse nerves (*Figure 7A*), due to their proper trafficking to or maintenance at synaptic sites. However, in *lrp4^dalek* mutants, Bruchpilot improperly accumulated in the transverse nerves (*Figure 7B*). This kind of accumulation is rarely observed in wild type, but is also most notably associated with loss of SRPK79D (*Figure 7C*), a conserved serine-arginine protein kinase that localizes to NMJ terminals and negatively regulates premature active zone assembly before Bruchpilot reaches the fly NMJ (*Johnson et al., 2009*; *Nieratschker et al., 2009*). In both *lrp4* and *srpk79D* mutants, Brp accumulation was not accompanied by focal accumulations of Synaptotagmin I, indicating that axonal trafficking is not generally impaired (*Figure 7A–C*) (*Gindhart et al., 1998*; *Johnson et al., 2009*; *Nieratschker et al., 2009*). Because of the similarity in the transverse nerve phenotypes and the role of SRPK79D at peripheral synapses, we hypothesized that LRP4 and SRPK79D could operate together in the CNS to regulate synapse number.

As SRPK79D antibodies are not available, we utilized Venus-tagged SRPK79D transgenes to examine CNS localization. When expressed only in VA1v ORNs, venus-SRPK79D localized to axon terminals and overlapped with Brp-Short, demonstrating localization with and adjacent to CNS active zones (*Figure 7D*). This was reminiscent of LRP4-HA localization in ORNs (*Figure 1F*) so we turned to proExM to more precisely assess the spatial relationship between SRPK79D and LRP4. In the ORNs of expanded individual glomeruli, LRP4-HA and venus-SRPK79D exhibited coincident and adjacent localization (*Figure 7E*). However, SRPK79D was expressed more broadly throughout ORNs, suggesting that only a subset of SRPK79D colocalizes with LRP4. This may indicate both LRP4-dependent and -independent roles for SRPK79D. We also examined this synaptic localization in *lrp4^dalek* mutants: loss of *lrp4* reduced synaptic SRPK79D levels by ~50% (*Figure 7F–H*). This reduction was specific for SRPK79D, as the staining for other markers, like the general neuropil label N-Cadherin, was unaffected (*Figure 7F–H*). These results demonstrate that LRP4 is necessary for the proper localization and / or expression of SRPK79D and suggest that SRPK79D might act downstream of LRP4 to regulate synapse number.

Due to their spatial proximity, we next employed proximity ligation assays (PLA) to determine whether LRP4 and SRPK79D are spatially close enough to interact. PLA uses oligonucleotides conjugated to secondary antibodies (*Greenwood et al., 2015*; *Söderberg et al., 2006*): if the epitopes are sufficiently close (30–40 nm), the oligonucleotides can be ligated together and detected using a fluorescent probe. The result can be observed using confocal microscopy and preserve, to a high degree, the spatial localization of the proteins involved. PLA has been used to examine protein-protein interactions at the NMJ (*Wang et al., 2015*) but not, to our knowledge, in the CNS. To examine this, we co-expressed venus-SRPK79D and LRP4-HA in all ORNs using *pebbled-GAL4* (*Sweeney et al., 2007*), stained both targets with oligonucleotide-conjugated secondary antibodies and performed PLA assays (*Figure 7I–J* and *Figure 7—figure supplement 1*). As expected, both proteins localize to the axon terminals of ORNs. When either is expressed singularly (*Figure 7—*

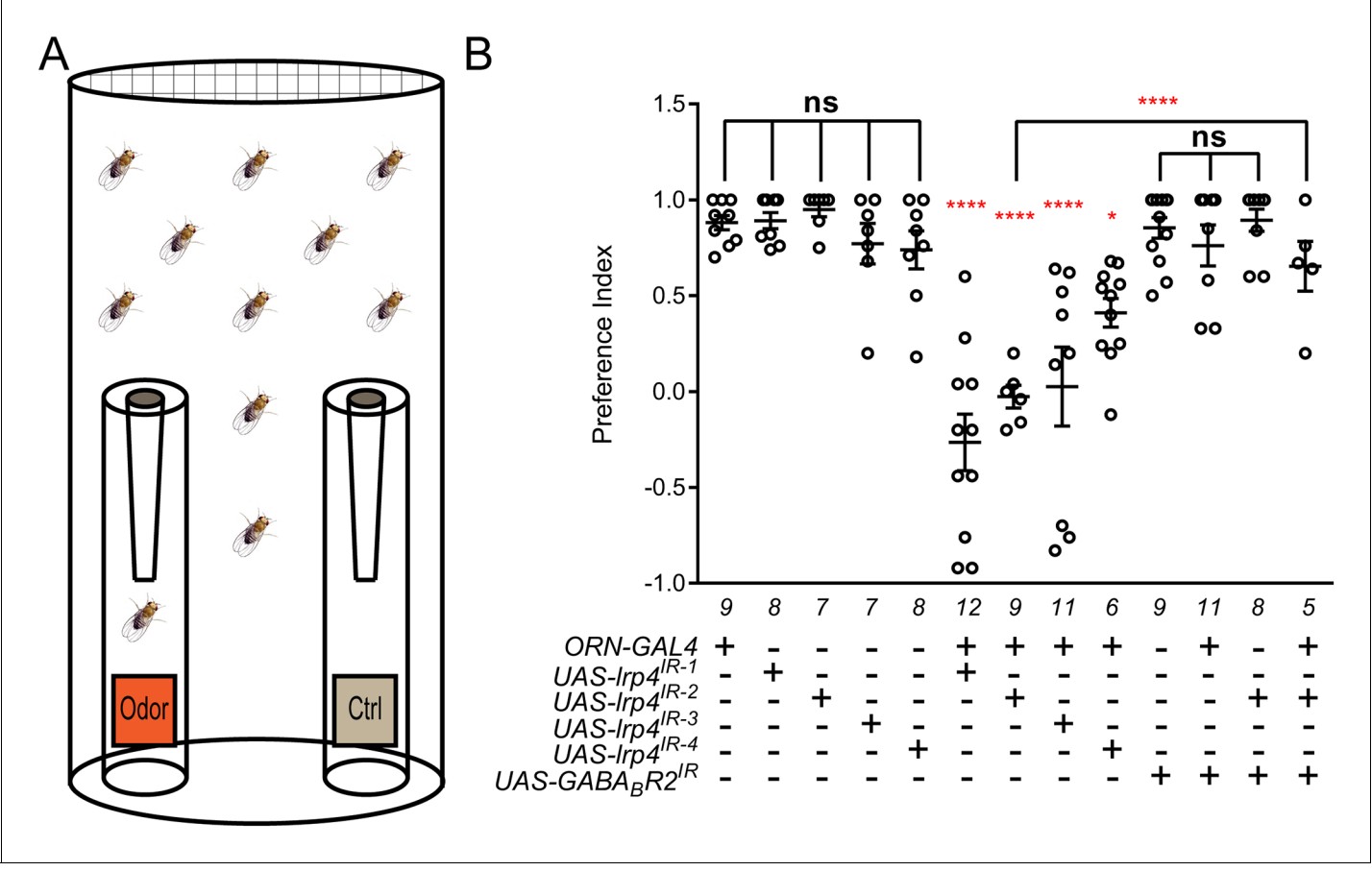

**Figure 6.** Loss of presynaptic LRP4 abolishes olfactory attraction behavior. (**A**) Cartoon of the olfactory trap. (**B**) Quantification of preference index [(# of flies in odor vial – # of flies in control vial) / total # of flies] between apple cider vinegar (odor) and water (ctrl). Genotypes are indicated below. Control flies with only a *GAL4* or *UAS-RNAi* transgene demonstrate high preference for the attractive odorant in apple cider vinegar. Flies expressing *lrp4* RNAi in ORNs have this attraction abrogated. Flies expressing RNAi against GABA$_B$R2 in ORNs still display robust attractive behavior while concurrent expression with *lrp4* knockdown largely suppresses the loss of attractive behavior. To ensure an equivalent number of transgenes in each genotype, *UAS-mCD8-GFP* was included (not listed) to control for potential transgenic dilution. ****$p<0.0001$; **$p<0.01$; *$p<0.05$; ns, not significant. Statistical comparisons (one-way ANOVA with correction for multiple comparisons) are with control unless otherwise noted. Error bars represent mean ± s.e.m. *n* (cohorts of 25 flies tested) is noted at the bottom of each column.

figure supplement 1A–B) or the probes are not added (*Figure 7I*), no PLA signal is observed. However, in the presence of both transgenes and the appropriate probes (*Figure 7J* and *Figure 7—figure supplement 1C–D*), we detected positive signal indicating that the proteins were close enough to interact. The PLA signal represented a subset of LRP4 or SRPK79D staining patterns, suggesting that there are roles independent of the other for each protein. Taken together, this data suggests that LRP4 interacts with SRPK79D to maintain SRPK79D localization at the synapse.

## SRPK79D overexpression suppresses LRP4 phenotypes

The interaction with, and reliance on LRP4 for synaptic SRPK79D localization suggested that the two function together. If so, we would expect that the two would display phenotypic similarity and interact in the same genetic pathway. We observed phenotypic similarity in larval nerves (*Figure 7A–C*), but we further sought to study this at CNS synapses. To test the interactions between LRP4 and SRPK79D with respect to effects of synapse number, we conducted loss-of-function, genetic interaction, and genetic epistasis experiments between genetic perturbations of both. First, reducing *srpk79D* function presynaptically using an established RNAi (*Johnson et al., 2009*) expressed in VA1v ORNs resulted in a 15% reduction in the number of Brp-Short puncta compared to control

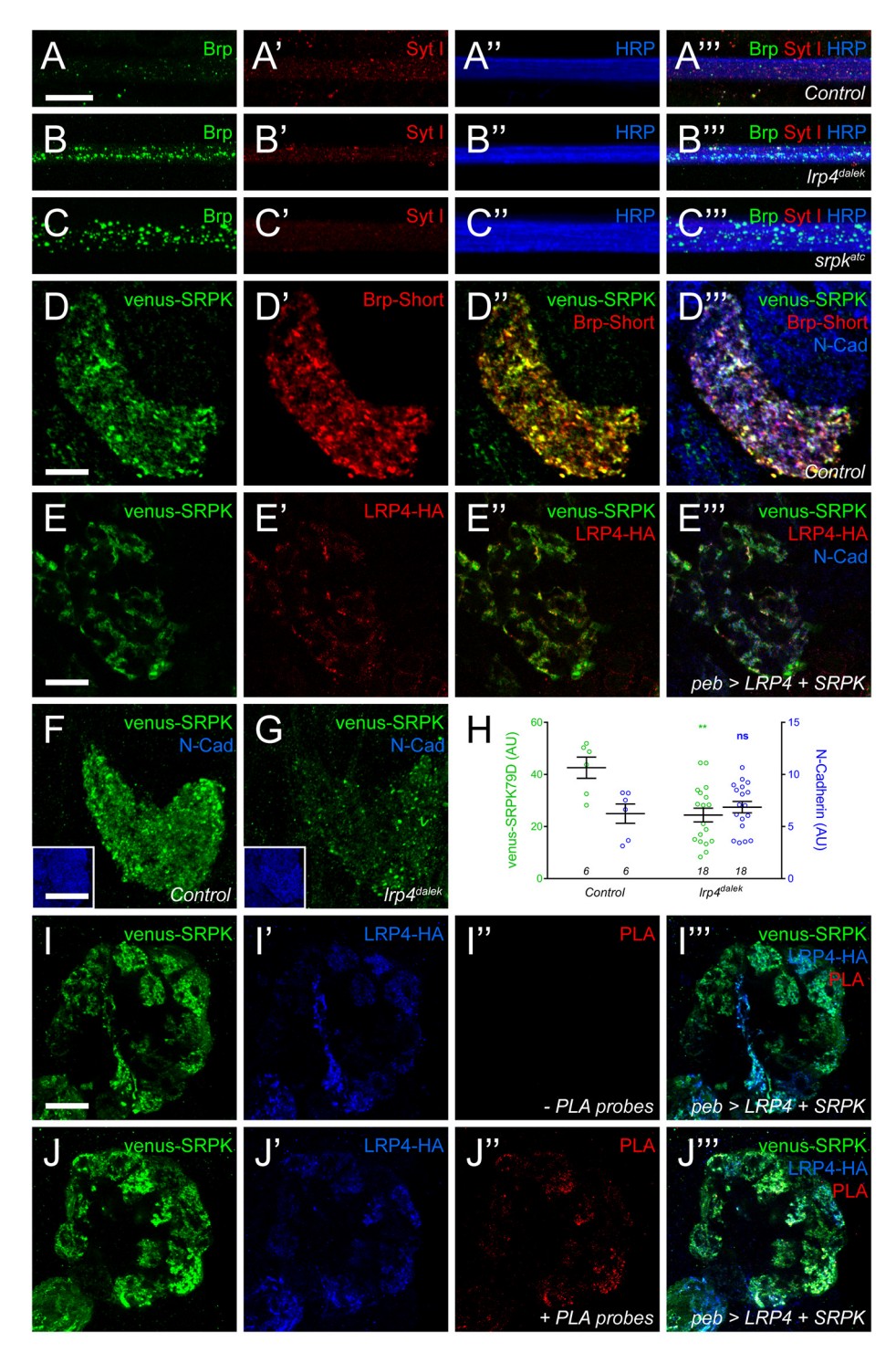

**Figure 7.** LRP4 is required for normal synaptic SRPK79D localization in the CNS. (**A–C**) Representative images of larval transverse nerves stained with antibodies to Bruchpilot (Brp, green), Synaptotagmin I (Syt I, red), and HRP (blue). Loss of *lrp4* (B, *lrp4^dalek^*) and *srpk79d* (C, *srpk^atc^*) result in improper axonal accumulations of Brp. This is not a general trafficking defect, as Syt I is absent from focal accumulations. (**D**) Representative high magnification confocal slice of VA1v ORNs expressing Brp-Short-mStraw and venus-SRPK79D and stained with antibodies to mStraw (red), GFP (green), and N-Cadherin (blue). SRPK79D largely colocalized with Brp-Short-mStraw but Brp-Short-positive / SRPK79D-negative and Brp-Short-negative / SRPK79D-positive puncta were also observed (**D''**). (**E**) Representative confocal slice within a single antennal lobe glomerulus of a brain expressing venus-SRPK79D and

*Figure 7 continued on next page*

*Figure 7 continued*

LRP4-HA in all ORNs, processed for proExM, and stained with antibodies to venus (green), HA (red), and N-Cadherin (blue). Distinct regions of overlap between venus-SRPK79D and LRP4-HA (E″) are observed, though this represents a subset of venus-SRPK79D localization. (F–G) Representative high magnification confocal maximum intensity projections of VA1v ORN axon terminals expressing venus-SRPK79D in control (F) and *lrp4*^*dalek*^ (G) backgrounds and stained with antibodies to GFP (green) and N-Cadherin (blue, inset). Loss of *lrp4* results in reduced synaptic SRPK79D. (H) Quantification of venus-SRPK79D (green, left axis) and N-Cadherin fluorescence (blue, right axis). SRPK79D fluorescence is markedly reduced in *lrp4*^*dalek*^ animals, but N-Cadherin staining is unaffected, demonstrating specificity. (I–J) Representative high magnification single confocal slices of the antennal lobe where all ORNs are expressing venus-SRPK79D and LRP4-HA via the *pebbled-GAL4* driver and the brains subsequently processed using proximity ligation assays to determine whether the two proteins were close enough to interact. The brains were stained with antibodies to venus (green) and HA (blue) and PLA-specific probes (red) to detect proximity ligation events. When PLA-specific probes are not added, no signal is observed (I″) but when present, positive PLA signal (J″) indicates close physical proximity between LRP4-HA and venus-SRPK79D. Positive PLA signal represents a subset of SRPK79D or LRP4 expression, as in (E). **p<0.01; ns, not significant. Statistical comparisons (one-way ANOVA with correction for multiple comparisons) are with control unless otherwise noted. Error bars represent mean ± s.e.m. *n* (antennal lobes) is noted at the bottom of each column. Scale bars = 10 μm (A–D,I–J), 25 μm (E), 20 μm (F–G), 33 μm (F-G insets).

The following figure supplement is available for figure 7:

**Figure supplement 1.** Proximity ligation assays reveal LRP4 and SRPK79D interactions.

---

(*Figure 8A–B,E*). Thus, SRPK79D is required for normal CNS synapse number. We further sought to understand if LRP4 and SRPK79D interacted genetically. To examine this, we performed a transheterozygote genetic interaction assay. When single copies of either *lrp4* or *srpk79D* were removed, there was no evident phenotype (*Figure 8—figure supplement 1B–C,E*). However, when one copy of each was concurrently removed, we observed a significant reduction in Brp-Short puncta (*Figure 8—figure supplement 1D–E*). This suggests that the two function in the same genetic pathway and may work together to ensure proper synapse number. Given the reduction in synaptic SRPK79D present in *lrp4* mutants, we examined whether these reduced SRPK79D levels are the root cause of its synapse reduction. We overexpressed SRPK79D in presynaptic ORNs either in control or *lrp4*^*dalek*^ mutant backgrounds. Presynaptic overexpression of SRPK79D in VA1v ORNs partially suppressed the synaptic phenotype associated with the *lrp4*^*dalek*^ mutation, resulting in 92% of the normal number of synapses (*Figure 8A,C–E*), whereas overexpression of SRPK79D in a wild-type background had no effect (*Figure 8E*). Finally, we sought to determine whether *srpk79D* was required for the increase in Brp-Short puncta associated with LRP4 overexpression (*Figure 2H*). When LRP4 was overexpressed concurrently with *srpk79D* RNAi, the phenotype resembled that of *srpk79D* RNAi alone (*Figure 8E*). This suggests that LRP4 requires SRPK79D to mediate its overexpression phenotype, likely by functioning through SRPK79D to increase the number of synapses. Combined, these indicate that LRP4 and SRPK79D closely interact presynaptically in the same genetic pathway to ensure the proper number of excitatory synapses.

In light of the synapse number defects, we also examined the functional consequences of *srpk79D* perturbation on olfactory behavior. Flies expressing *srpk79D* RNAi in all ORNs demonstrated a nearly complete abrogation of attraction behavior (*Figure 8F*) that was indistinguishable from the *lrp4* RNAi phenotype. In light of the suppression of the synapse number phenotype, we also examined whether SRPK79D overexpression could suppress the *lrp4* loss-of-function behavioral phenotype. Control flies bearing the pan-ORN *pebbled-GAL4* or the SRPK79D overexpression transgene alone exhibited strong attraction towards apple cider vinegar (*Figure 8F*). Further, SRPK79D overexpression in all ORNs did not affect this robust attraction. Driving both SRPK79D overexpression and *lrp4* RNAi in all ORNs, however, resulted in a partial suppression of the behavioral phenotype associated with *lrp4* RNAi (*Figure 8F*). As the synaptic level of SRPK79D is positively regulated by LRP4 and SRPK79D overexpression suppresses the morphological and functional phenotypes associated with *lrp4* loss-of-function, SRPK79D is likely a key downstream effector of LRP4 in regulating synapse number and thus, normal olfactory attraction behavior.

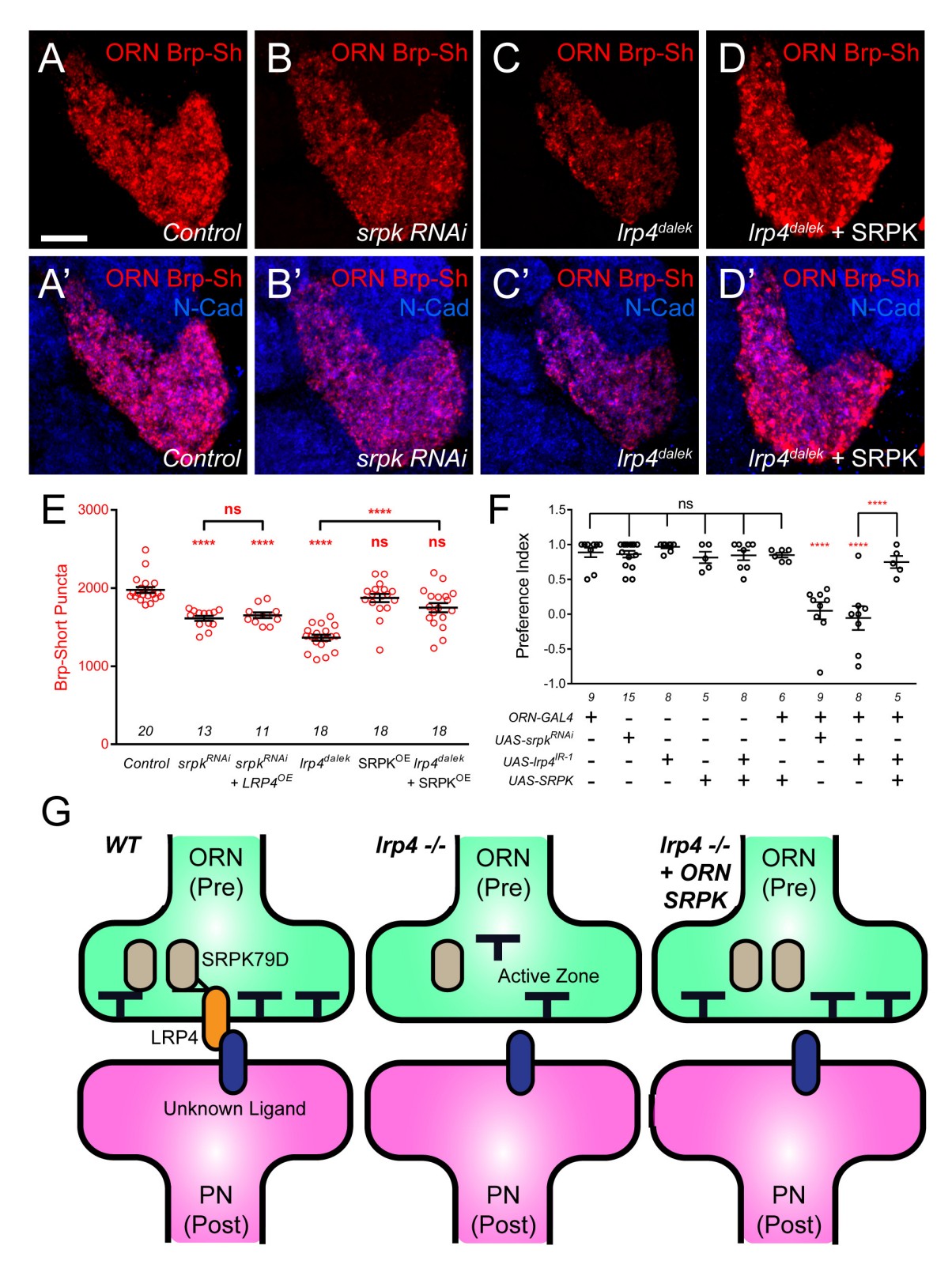

**Figure 8.** SRPK79D and LRP4 genetically interact to control synapse morphology and function. (**A–D**) Representative high magnification confocal maximum intensity projections of VA1v ORN axon terminals in males expressing Brp-Short-mStraw and stained with antibodies to mStraw (red) and N-Cadherin (blue). Presynaptic RNAi against *srpk79D* (*srpk RNAi*) reduces the number of puncta, but less so than loss of *lrp4* (*lrp4^dalek^*). Presynaptic overexpression of SRPK79D in an *lrp4^dalek^* background (*lrp4^dalek^ + SRPK*) restores puncta number to control levels. (**E**) Quantification of Brp-Short-

*Figure 8 continued on next page*

*Figure 8 continued*

mStraw puncta. Note that overexpression of SRPK79D in an otherwise wild-type background has no gain-of-function effects on puncta number. Further *srpk79D* function is needed to enable the LRP4 overexpression-induced increase in synaptic puncta number. *n* (antennal lobes) is noted at the bottom of each column. (F) Quantification of preference index in the olfactory trap assay. Flies overexpressing SRPK79D in ORNs show strong attractive behavior, while ORNs expressing RNAi against *lrp4* or *srpk79D* abrogate attraction to apple cider vinegar. This phenotype can be suppressed by concurrent overexpression of SRPK79D. *UAS-mCD8-GFP* (not listed) was used to ensure equivalent numbers of transgenes in each genotype. *n* (cohorts of 25 flies tested) is noted at the bottom of each column. (G) A model for LRP4 function at olfactory synapses. At wild-type axon terminals, LRP4 in presynaptic ORNs (orange) interacts with a putative postsynaptic partner (blue), resulting in SRPK79D (beige) retention at the terminal and a full complement of active zones (black T). Here, the putative ligand is depicted as having a postsynaptic PN source, but alternate sources (such as glia or local interneurons) are also possible. In the absence of LRP4, less synaptic SRPK79D is present and active zone number is reduced. The size of the terminal itself does not change but the synapse number (i.e., number of active zones) within that terminal space is reduced. Further, T-bar defects like a floating T-bar can also be seen. SRPK79D overexpression in an *lrp4* mutant restores synaptic SRPK79D and active zone number, despite the absence of LRP4. Thus, the LRP4 largely functions in synaptic organization through downstream SRPK79D. ****p<0.0001; ***p<0.001; ns, not significant. Statistical comparisons (one-way ANOVA with correction for multiple comparisons) are with control unless otherwise noted. Error bars represent mean ± s.e.m. *n* (antennal lobes for E, cohorts of 25 flies tested for F) is noted at the bottom of each column. Scale bars = 10 μm.

The following figure supplement is available for figure 8:

**Figure supplement 1.** *lrp4* and *srpk79D* interact genetically to control Brp-Short puncta number.

## Discussion

Understanding how synaptic organizers regulate the number and function of synapses in the CNS is a central goal of molecular neurobiology. This study identifies LRP4 as a synaptic protein whose expression may be preferential for excitatory neurons in the *Drosophila* CNS (*Figure 1*). Though well-known as the postsynaptic agrin receptor at the mouse NMJ (*Kim et al., 2008*; *Zhang et al., 2008*), here we describe an agrin-independent, presynaptic role for LRP4. In the *Drosophila* CNS, LRP4 functions presynaptically to regulate the number of active zones in presynaptic ORNs (*Figures 2–3*) and acetylcholine receptor clusters in the PNs postsynaptic to those ORNs (*Figure 2*). Moreover, LRP4 also controls the morphology of individual active zones: *lrp4* mutant T-bars exhibit striking defects in patterning and biogenesis (*Figure 3*). These defects are specific for excitatory neurons, as inhibitory neuron synapses in the antennal lobe remain unaffected (*Figure 4*). Overexpression of LRP4, however, can increase synapse number cell autonomously in both excitatory and inhibitory neurons (*Figures 2–4*), suggesting that both share common mechanisms for synapse addition. The role for LRP4 further extends to higher order olfactory neuropil in the lateral horn (*Figure 5*), suggesting that it may serve a general role in synaptic organization. Underscoring the functional importance of LRP4, its perturbation in excitatory ORNs abrogated olfactory attraction behavior (*Figure 6*). The suppression of the behavioral phenotype by reducing presynaptic inhibition onto ORNs further suggests that a proper level of excitatory drive is important for functional circuit output. To mediate both morphological and behavioral effects, LRP4 likely functions through SRPK79D, a conserved SR-protein kinase whose loss-of-function phenotypes resemble those of *lrp4* (*Figures 7–8*), whose synaptic localization depends on LRP4 (*Figure 7*), who interacts genetically with and is physically in proximity to LRP4 (*Figures 7–8*), and whose overexpression suppresses the phenotypes associated with loss of *lrp4* (*Figure 8*).

### LRP4 as a synaptic regulator that distinguishes excitatory from inhibitory presynaptic terminals

Coordination of excitation and inhibition is critical to proper circuit function. Imbalances in excitation and inhibition lead to epileptic states (*Badawy et al., 2012*) and social dysfunction (*Yizhar et al., 2011*), and may also underlie many autism spectrum disorders (*Mullins et al., 2016*; *Nelson and Valakh, 2015*). The mechanisms that maintain this balance are incompletely understood, though likely involve multiple aspects including the number of each type of neuron, their firing rates, release probabilities, synaptic strength, and neurotransmitter receptor sensitivities. Such regulation likely requires distinguishing excitatory from inhibitory neurons at both pre- and postsynaptic levels. Excitatory and inhibitory synapses are identified postsynaptically by distinct neurotransmitter receptor, scaffolding protein, and adhesion molecule repertoires (*Craig and Kang, 2007*; *Sheng and Kim,*

*2011*; *Ziv and Fisher-Lavie, 2014*). Postsynaptic factors like Neuroligin 2 (*Graf et al., 2004*), Gephyrin (*Choii and Ko, 2015*), and Slitrk3 (*Takahashi et al., 2012*) organize inhibitory GABAergic synapses while LRRTMs organize excitatory synapses (*Siddiqui et al., 2013*; *de Wit et al., 2009*, *2013*). Thus, postsynaptic regulation can occur by differential modulation of these factors. Little is known, however, about the presynaptic identifiers of excitatory versus inhibitory neurons. Recent work identified Punctin / MADD-4 as a determinant of excitatory versus inhibitory neuromuscular synapses in *C. elegans*, though as a secreted factor that functions via postsynaptic interaction (*Maro et al., 2015*; *Pinan-Lucarré et al., 2014*; *Tu et al., 2015*). Further, Glypican4 can localize to excitatory presynaptic terminals and interact with LRRTM4 (*de Wit et al., 2013*) but its synaptogenic activity is also provided by astrocytes (*Allen et al., 2012*) and thus is not neuronal specific. Proteomic comparisons (*Biesemann et al., 2014*; *Boyken et al., 2013*) suggest few differences beyond those pertaining to neurotransmitter synthesis enzymes and transporters. But these components may not be sufficient to distinguish presynaptic excitatory from inhibitory neurons. In the *Drosophila* olfactory system, for example, glutamate can be inhibitory when its postsynaptic partners express glutamate-gated chloride channels (*Liu and Wilson, 2013*). This suggests that pre- and postsynaptic regulators may exist to distinguish excitatory and inhibitory synapses, though it is unclear what those presynaptic regulators might be.

Our data suggests that LRP4 may be a candidate presynaptic organizer specific for excitatory connections. LRP4 is expressed in a subset of excitatory cholinergic neurons, excluded from inhibitory GABAergic neurons, and expressed in a subset of glutamatergic neurons that may be excitatory or inhibitory (*Figure 1*). Though we cannot rule out inhibitory neuron expression in the case of the glutamatergic subset, the phenotypes associated with LRP4 perturbation are consistent with an excitatory neuron-specific role. Thus, LRP4 may not only serve an identifying role at excitatory synapses, but also a functional one. Loss of *lrp4* results in fewer excitatory synapses but has no effect on inhibitory synapses. However, both excitatory and inhibitory neurons show increased synapse number with *lrp4* overexpression (*Figures 2* and *4–5*). This shared competency suggests that both neurons contain machinery that can be engaged downstream of LRP4 (or the cell surface) to add synapses. Thus, proteins like LRP4 may represent identifiers of excitatory or inhibitory terminals that function by engaging common mechanisms to add synapses.

## LRP4 function across evolution

At the mouse NMJ, LRP4 is the well-established postsynaptic receptor for motoneuron-derived Agrin (*Kim et al., 2008*; *Zhang et al., 2008*, *2011*) and regulates synapse formation (*Weatherbee et al., 2006*) and maintenance (*Barik et al., 2014*). However, additional roles for LRP4 exist at the level of the presynaptic motoneuron. A retrograde signal composed of LRP4 from the postsynaptic muscle interacts with an unknown receptor in the motoneuron (*Yumoto et al., 2012*) to regulate presynaptic differentiation. Thus, at the mouse NMJ, postsynaptic LRP4 has both cell-autonomous and non-cell autonomous roles. In addition, presynaptic LRP4 has been implicated to regulate acetylcholine receptor clustering via MMP-mediated proteolytic cleavage (*Wu et al., 2012*).

In the mouse CNS, LRP4 regulates synaptic physiology (*Gomez et al., 2014*; *Pohlkamp et al., 2015*), learning and memory, fear conditioning, and CA1 spine density (*Gomez et al., 2014*). Though CNS LRP4 most commonly associates with postsynaptic densities (*Tian et al., 2006*), it also fractionates with synaptophysin-positive membranes (*Gomez et al., 2014*). Indeed, the observed CNS phenotypes have not been localized to a particular pool of LRP4. Our identification of *Drosophila* LRP4 as a key player in CNS synaptogenesis, however, posits a cell-autonomous presynaptic role. While we cannot rule out an additional, perhaps concurrent, postsynaptic role, our work is the first to demonstrate clear cell-autonomous presynaptic functions for LRP4. Indeed, LRP4 is expressed in PNs and may localize to PN dendrites within the antennal lobe (*Figure 1—figure supplement 2*). In such a case, it could function either presynaptically, at dendrodendritic presynapses (*Rybak et al., 2016*; *Tobin et al., 2017*) or as a postsynaptic factor. Moreover, as the *Drosophila* genome lacks clear Agrin and MuSK homologs, this suggests a synaptic function of LRP4 that evolutionarily precedes Agrin and MuSK recruitment to vertebrate NMJ synaptogenesis.

It remains open whether this presynaptic function is conserved in the mammalian CNS and, if so, what signal LRP4 receives. In *Drosophila*, the signal cannot be Agrin and in the mammalian CNS, Agrin is not essential for CNS synapse formation (*Daniels, 2012*). Thus, the Agrin-independence of CNS LRP4 may be conserved across systems. Moreover, our finding that LRP4 promotes excitatory,

but not inhibitory, synapse formation and function is consistent with reduced excitatory but normal inhibitory input in hippocampal CA1 neurons of *lrp4* mutant mice (*Gomez et al., 2014*). Moreover, we find that LRP4 in the *Drosophila* CNS functions through the SR-protein kinase SRPK79D. Impaired *srpk79D* function reduces synapse number and overexpression can suppress the functional and morphological defects associated with *lrp4* loss (*Figures 7–8*). This kinase is evolutionarily conserved (*Johnson et al., 2009*) and the three mammalian homologues (*Zhou and Fu, 2013*) are widely expressed in the mouse brain (*Lein et al., 2007*), including in the hippocampus. From yeast to human, SRPKs regulate spliceosome assembly and gene expression (*Zhou and Fu, 2013*) but have not been studied in mammalian synapse formation. It will be interesting to test if these kinases also function in the mammalian CNS. Combined, however, these commonalities suggest a basic conservation between invertebrate and vertebrate systems for future study.

### Connecting LRP4 and human disease

Recent work implicated LRP4 in both amyotrophic lateral sclerosis (ALS) and myasthenia gravis (MG), two debilitating motor disorders with a worldwide prevalence of ~1/5000. Distinct ALS and MG populations are seropositive for LRP4 autoantibodies (*Tsivgoulis et al., 2014*; *Tzartos et al., 2014*) and double seronegative for Agrin or MuSK, suggesting that seropositivity is not a byproduct of generalized NMJ breakdown. Further, injection of LRP4 function-blocking antibodies into mice recapitulates MG (*Shen et al., 2013*). Beyond peripheral symptoms, cognitive impairment (besides that as frontotemporal dementia) also occurs in a subset of ALS patients (*Ringholz et al., 2005*). Thus, understanding the roles of LRP4 in the peripheral and central nervous systems has marked clinical significance. Our identification of an evolutionarily conserved kinase, SRPK79D, as a downstream target of LRP4 signaling may offer a window into those roles. As SRPK79D overexpression suppresses the behavioral and the synaptic phenotypes of *lrp4* loss (*Figure 8*), if it functions similarly in the mammalian CNS, SRPKs could be a target for therapeutics. Further investigation of how LRP4 functions in the CNS will provide new insight not only into the cognitive aspects of these debilitating motor disorders, but also into the fundamental aspects of excitatory synapse formation.

## Materials and methods

### Generation of lrp4 CRISPR mutants

The *lrp4* mutation was designed following published methods (*Gratz et al., 2013*). Two *lrp4*-specific chimeric RNAs (chiRNA) were cloned into the pU6-BbsI-chiRNA vector as follows - A1, corresponding to an optimal PAM site 2 bp 5′ of the start ATG (using primers: 5′ CTTCGGCGAGTTTGTGTACA TGTC 3′ and 5′ AAACGACATGTACACAAACTCGCC 3′ with a phosphate at the 5′ end) and A2, corresponding to an optimal PAM site 34 bp 3′ of the TAG stop codon (using primers 5′ CTTCGAA TCGGTAAATGGTTTCAG 3′ and 5′ AAACCTGAAACCATTTACCGATTC 3′). Both the A1 and A2 chiRNA plasmids (250 ng / μL) and a pHsp70-Cas9 plasmid (500 ng / μL) were injected into MB03015 embryos (stock BL23835) to produce *lrp4* deletions. MB03015 flies bear a Minos-based Mi {ET1} insertion (*Bellen et al., 2011*) between exons 5 and 6 of the *lrp4* open reading frame; adults with the insertion are marked by expression of a GFP reporter in the eye. Successful events were screened for by the loss of GFP: as the PAM sites were distant from and flanking the insertion, loss of fluorescence likely indicated removal of the intervening sequences (the *lrp4* coding region). Five such lines (representing identical events) were recovered and homozygous viable stocks established: the allele was named *dalek* due to the 'extermination' of the *lrp4* gene, and in homage to the classic villains of 'Doctor Who'. Loss of *lrp4* was assessed using genomic DNA prepared from control and *lrp4*^dalek^ adults using the QIAgen DNeasy Blood and Tissue Kit (QIAgen, Valencia, CA). Genomic PCR bands corresponding to exon 2 (534 bp using primers 5′ TGTATTCCACGAACCTGGGTATG 3′ and 5′ CAAAATGCAGCGCCCATTGTT 3′) and the exon 7–8 junction (615 bp using primers 5′ AGTC TTGATGGTAGCAATAGGCAT 3′ and 5′ CTCTGGTAGATTTTGACACTG 3′) revealed the absence of both regions in *lrp4*^dalek^. The *lrp4*^dalek^ deletion was further confirmed by the presence of a 315 bp 'Flank' band (with some background bands present only with the *lrp4*^dalek^ deletion) representing the connection of sequences from the 5′ and 3′ UTRs (amplified by primers 5′ AACAGAATCGGAACAG-CAGTT 3′ and 5′ GAGCTTTAACAGGACACGTTT 3′) not present in control samples (see *Figure 1—*

*figure supplement 2B*). Finally, antibody staining (see below) revealed the elimination of LRP4 signal in the *lrp4*^dalek^ allele, suggesting the creation of a null allele.

## Cloning of LRP4 cDNA and transgene construction

An adult *Drosophila* cDNA library was made according to manufacturer's protocol using the GeneRacer Kit (ThermoFisher Scientific, Catalog #L150201, Waltham, MA). From the library, the *lrp4* cDNA was amplified using the forward primer 5' CACCATGTATTTGACAGCCTTT 3' and the reverse primer 5' TGTGATAGTCGAGAGCGT 3' (without the endogenous Stop codon) and cloned directly into the pENTR vector using the pENTR/D-TOPO Cloning Kit (ThermoFisher Scientific, Catalog #K240020, Waltham, MA). Complete cDNA clones were verified by sequencing. UAS-LRP4-HA was made by recombining pENTR-LRP4 with pUAST-attB-Gateway-3xFLAG-3xHA[29] via LR clonase. The resultant pUAST-attB-LRP4-3xHA-3xFLAG was transformed into the ΦC31 landing site 86Fb on the 3^rd^ chromosome using standard methods.

## Production of LRP4 antibodies

Custom antibodies were made by Pierce Custom Services (ThermoFisher, Rockford, IL) against the C-NKRNSRGSSRSVLTFSNPN peptide corresponding to residues 1921–1939 of the intracellular side of LRP4. Rat antisera were Ig-purified and then used at a dilution of 1:200 on adult brains. The specificity of the antibody was verified by the absence of signal in the *lrp4*^dalek^ mutant.

## Alignment of LRP4 homologues

The *Drosophila melanogaster* (CG8909; accession AAF48538.1), *Mus musculus* (accession NP_766256.3), and *Homo sapiens* (accession NP_002325.2) LRP4 sequences were obtained from NCBI. CLUSTALW alignment was performed using PSI/T-Coffee for transmembrane proteins (http://tcoffee.crg.cat/apps/tcoffee/do:tmcoffee) and expressed graphically using ESPript3.0 (http://espript.ibcp.fr/ESPript/ESPript/).

## *Drosophila* stocks and transgenic strains

All controls, stocks, and crosses were raised at 25°C. Mutants and transgenes were maintained over balancer chromosomes to enable selection in adult or larval stages. The GMR90B08-GAL4 (*Pfeiffer et al., 2008*) line was used to examine *lrp4* expression (referred to as *lrp4*-GAL4). Four UAS-RNAi lines against differing regions of *lrp4* were also identified: *UAS-lrp4-RNAi 1* (v29900, Vienna Drosophila Resource Center), *UAS-lrp4-RNAi 2* (v108629, Vienna Drosophila Resource Center), *UAS-lrp4-RNAi 3* (JF01570, Harvard TRiP Collection), *UAS-lrp4-RNAi 4* (JF01632, Harvard TRiP Collection). The following GAL4 lines enabled tissue-specific expression: *Or47b-GAL4* (VA1v ORNs) (*Vosshall et al., 2000*), *Or67d-GAL4* (DA1 ORNs) (*Kurtovic et al., 2007*), *Or88a-GAL4* (VA1d ORNs) (*Vosshall et al., 2000*), *AM29-GAL4* (DL4 and DM6 ORNs) (*Endo et al., 2007*), *Mz19-GAL4* (DA1, VA1d, DC3 PNs) (*Jefferis et al., 2004*), *Mz699-GAL4* (inhibitory projection neurons that project to the lateral horn) (*Lai et al., 2008*; *Liang et al., 2013*), *GAD1-GAL4* (GABAergic inhibitory neurons) (*Ng et al., 2002*), *pebbled-GAL4* (all ORNs) (*Sweeney et al., 2007*). The following UAS transgenic lines were used as either reporters or to alter gene function: UAS-Syt-HA (*Robinson et al., 2002*), *UAS-Brp-Short-mStraw* (*Fouquet et al., 2009*), *UAS-DSyd1-GFP* (*Owald et al., 2010*), *UAS-Dα7-GFP* (*Leiss et al., 2009*), *UAS-mCD8-GFP* (*Lee and Luo, 1999*), *UAS-3xHA-mtdT* (*Potter et al., 2010*), *UAS-FRT-Stop-FRT-mCD8-GFP* (*Hong et al., 2009*), *UAS-Dcr2* (*Dietzl et al., 2007*), *UAS-GABA_BR2-RNAi* (*Root et al., 2008*), *UAS-srpk79D-RNAi* (*Johnson et al., 2009*), *UAS-venus-SRPK79D-#28* (*Johnson et al., 2009*), *UAS-venus-SRPK79D-#1A* (*Johnson et al., 2009*). Intersectional analyses were done using the *eyFLP*^3.5^ construct (*Chotard et al., 2005*) which expresses FLP in ORNs, but not PNs and *GH146-FLP* (*Hong et al., 2009*), which expresses in 2/3 of all olfactory PNs but not ORNs. The *srpk79D*^atc^ allele (*Johnson et al., 2009*) was used to remove *srpk79D* function.

## Immunocytochemistry

Adult brains were dissected at 10 days post eclosion as previously described (*Mosca and Luo, 2014*; *Wu and Luo, 2006*). Third instar larvae were dissected as previously described (*Mosca and Schwarz, 2010*). The following primary antibodies were used: mouse antibody to Bruchpilot (1:40,

DSHB, Catalog #mAbnc82, Iowa City, IA) (*Laissue et al., 1999*), rabbit antibody to Synaptotagmin I (1:4000) (*Mackler et al., 2002*), rat antibody to N-Cadherin (1:40, DSHB, Catalog #mAbDN-EX #8, Iowa City, IA) (*Iwai et al., 1997*), rat antibody to HA (1:100, Roche, Catalog #11867423001, Basel, Switzerland), mouse antibody to choline acetyltransferase (ChAT) (1:100, DSHB, Catalog #mAb-ChAT4B1, Iowa City, IA) (*Takagawa and Salvaterra, 1996*), mouse antibody to ELAV (DSHB, mAb9F8A9, 1:100) (*O'Neill et al., 1994*), rabbit antibody to GABA (1:200, Sigma-Aldrich, Catalog #A2052, St. Louis, MO), mouse antibody to Repo (1:100, DSHB, Catalog #mAb8D12, Iowa City, IA) (*Alfonso and Jones, 2002*), rabbit antibody to vGlut (1:500) (*Daniels et al., 2008*), rabbit antibody to dsRed (1:250, Clontech, Catalog #632496, Mountain View, CA), chicken antibody to GFP (1:1000, Aves Labs, Catalog #GFP-1020, Tigard, OR), Alexa647-conjugated goat antibody to HRP (1:100, Jackson ImmunoResearch, Catalog #123-605-021, West Grove, PA). Alexa488-, Alexa568-, and Alexa647-conjugated secondary antibodies were used at 1:250 (ThermoFisher Scientific and Jackson ImmunoResearch, Various Catalog #s). CF633-conjugated secondary antibodies were used at 1:250 (Biotium). FITC-conjugated secondary antibodies were used at 1:200 (Jackson ImmunoResearch, Catalog #703-095-155, West Grove, PA).

## Proximity ligation assay

Brains were processed as described and stained using rabbit anti-GFP antibodies at 1:500 (Thermo-Fisher Scientific, Catalog #A-11122, Waltham, MA) with FITC-conjugated secondary antibodies and mouse anti-HA antibodies at 1:250 (Sigma-Aldrich, Catalog #A2095, St. Louis, MO) with Alexa647-conjugated secondary antibodies, leaving the red channel open. For PLA, we used the DuoLink Mouse Rabbit in situ PLA kit (Sigma-Aldrich, Catalog #DUO92101, St. Louis, MO). Following the last wash after secondary antibody incubation, the brains were incubated in the anti-mouse and / or anti-rabbit PLA probes at a 1:5 dilution for 2 hr at 37°C. Brains were then washed thrice for 10' each with Wash Buffer A, and incubated in Ligation solution (1:40 ligase in ligation buffer) for 1 hr at 37°C. Brains were washed in Wash buffer A for three times at 10' each and then incubated in Amplification solution (1:80 dilution of polymerase in Amplification buffer) for 2 hr at 37°C. Finally, brains were washed three times for 10' each in Wash Buffer B, and incubated in SlowFade overnight before mounting. Controls without Probes went through the identical process as those with probes, but with water substituted for the probes themselves in the first PLA step. Brains were imaged as described via confocal microscopy.

## Imaging, synaptic quantification and image processing

All images were obtained using a Zeiss LSM510 Meta laser-scanning confocal microscope (Carl Zeiss, Oberkochen, Germany) using either a 40 × 1.4 NA PlanApo or a 63 × 1.4 NA PlanApo lens. Images of synaptic puncta (Brp-Short-mStraw or Dα7-GFP) and neurite membrane (mCD8-GFP, 3xHA-mTDT) were imaged, processed and quantified as previously described (*Mosca and Luo, 2014*) with the following adjustments: images of synaptic puncta in the lateral horn (Mz19-GAL4, Mz699-GAL4, *Figure 5*, *Figure 5—figure supplement 1*) were imaged at 63X, with an optical zoom of 2. Mz19 and Mz699 images were processed with a spot size of 0.6 μm and neurite volume calculated with a smoothing of 0.2 μm and a local contrast of 0.5 μm.

Images were processed and figures prepared using Adobe Photoshop CS4 and Adobe Illustrator CS4 (Adobe Systems, San Jose, CA). For antibody staining comparisons between genotypes, samples were imaged and processed under identical conditions. Fluorescence intensity was measured with ImageJ (NIH, Bethesda, MD).

## Electron microscopy

Transmission electron microscopy was performed on 10 day old adult control and *lrp4dalek* male brains as previously described (*Mosca and Luo, 2014*). Putative ORN terminals were identified based on morphology (*Rybak et al., 2016*; *Tobin et al., 2017*) and quantified as described (*Mosca and Luo, 2014*). Terminal perimeter was measured using ImageJ (NIH, Bethesda, MD) and used to calculate T-bar density. All quantification was done with the user blind to the genotype.

## Expansion microscopy

Protein retention expansion microscopy (*Tillberg et al., 2016*) was modified for use with *Drosophila* brain tissue. Fixed and antibody-labeled brains were treated with 100 μg / mL acryloyl-X, SE (ThermoFisher Scientific, Catalog #A20770, Waltham, MA) overnight at room temperature and then embedded in polyelectrolyte gel for two hours at 37°C. Slices containing brains were excised from solidified polyelectrolyte gel and immersed in digestion buffer with 200 μg / mL Proteinase K (ThermoFisher Scientific, Catalog #AM2546, Waltham, MA) overnight at room temperature. Slices achieved maximum expansion after five washes with deionized water. Fully expanded gel slices were anchored to the bottom of a petri dish with 2% low melting point agarose. Confocal microscopy images were obtained on a Leica SP8 with a 25x water immersion objective (Leica Microsystems, Wetzlar Germany).

## Statistical analysis

Statistical analysis was completed using Prism 6.07 (GraphPad Software, Inc., La Jolla, CA). For representative datasets, the experimenter was blind to genotype during quantification and data analysis. Significance between two samples was determined using student's t-test. Significance amongst multiple samples was determined using one-way ANOVA with a Tukey's post-test to correct for multiple comparisons. Significance between two samples (for EM) was determined using a two-tailed student's t-test.

## Behavioral analyses

Olfactory trap assays were constructed as described (*Potter et al., 2010*). Flies were raised in a 12/12 light/dark incubator. For each cohort, 25 flies of the appropriate genotype were starved overnight in a 1% agar vial in complete darkness. They were anesthetized briefly on ice and transferred to the olfactory trap, which contained an experimental vial of apple cider vinegar (ACV: Safeway, Palo Alto, CA) and a control vial of water. Flies were then left in the trap for 16 hr in complete darkness before being quantified. Preference index was calculated as $(Flies_{ACV} - Flies_{Water}) / Flies_{Total}$.

## Genotypes

See *Supplementary file 1* for a listing of complete genotypes by figure panel.

# Acknowledgements

We thank Graeme Davis, Kate O'Connor-Giles, and Jing Wang for the kind gift of reagents, the Bloomington Stock Center for flies, and the Developmental Studies Hybridoma Bank (University of Iowa, maintained under the auspices of the NICHD) for antibodies. We acknowledge the Luo Lab and S Zosimus for fruitful discussions, and T Clandinin, K Shen, T Südhof, K Beier, N Berns, L Denardo Wilke, and J Lui for critical comments on the manuscript. We also acknowledge T Clandinin for supporting IEW. This work was supported by grants from the National Institutes of Health (NIH: K99/R00-DC013059 to TJM; R01 DC-005982 to LL; SIG 1S10RR02678001 to Stanford University to support the Electron Microscopy Core). LL is an Investigator of the Howard Hughes Medical Institute.

# Additional information

## Competing interests

LL: Reviewing editor, *eLife*. The other authors declare that no competing interests exist.

## Funding

| Funder | Grant reference number | Author |
| --- | --- | --- |
| National Institute on Deafness and Other Communication Disorders | R01-DC005982 | Liqun Luo |
| Howard Hughes Medical Institute | | Liqun Luo |

| National Institute on Deafness and Other Communication Disorders | R00-DC013059 | Timothy J Mosca |
| National Institute on Deafness and Other Communication Disorders | K99-DC013059 | Timothy J Mosca |

The funders had no role in study design, data collection and interpretation, or the decision to submit the work for publication.

## Author contributions

TJM, Conceptualization, Resources, Formal analysis, Funding acquisition, Validation, Investigation, Visualization, Methodology, Writing—original draft, Project administration, Writing—review and editing, TJM conceived of the project, performed experiments, analyzed data, contributed new reagents, and wrote the paper. TJM and IEW performed expansion microscopy. All authors critically reviewed the paper; DJL, Investigation, Writing—review and editing, DJL made the CRISPR mutation in lrp4 and performed transgenic injections. All authors critically reviewed the paper; IEW, Investigation, Methodology, Writing—review and editing, IEW adapted expansion microscopy for Drosophila. TJM and IEW performed expansion microscopy. All authors critically reviewed the paper; LL, Supervision, Funding acquisition, Project administration, Writing—review and editing, LL supervised the project. All authors critically reviewed the paper

## Author ORCIDs

Timothy J Mosca, http://orcid.org/0000-0003-3485-0719
Liqun Luo, http://orcid.org/0000-0001-5467-9264

## Additional files

### Supplementary files

• Supplementary file 1. Table of genotypes for all experimental conditions. Full genotype information for each image or analyzed condition. Notation (+; +; +; +) follows standard *Drosophila* genetics, corresponding to the X, 2nd, 3rd, and 4th chromosomes. Corresponding figure panels are listed as well for each genotype.

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
