## [Decision Letter]

Thank you for submitting your article "Presynaptic LRP4 Promotes Synapse Number and Function of Excitatory CNS Neurons" for consideration by *eLife*. Your article has been favorably evaluated two reviewers, one of whom, K VijayRaghavan (Senior Editor), is a member of our Board of Reviewing Editors. The following individual involved in review of your submission has agreed to reveal their identity: Rachel Wilson (Reviewer #1).

The reviewers have discussed the reviews with one another and the Reviewing Editor has drafted this decision to help you prepare a revised submission. As you can see the comments are readily addressable. We look forward to a revised manuscript which will be speedily accepted.

*Reviewer #1:*

This is an impressive study of the molecular organizers of presynaptic neurotransmitter release sites. This study uses a remarkably wide range of experimental approaches that beautifully exploit the *Drosophila* genetic toolkit. Mosca et al. begin by identifying the *Drosophila* homolog of LRP4. LRP4 has well-documented roles at mammalian neuromuscular junctions as a postsynaptic receptor organizer (and agrin receptor), and it clearly has roles at mammalian CNS synapses also, but its CNS roles are relatively poorly understood. Here, Mosca et al. show that it LRP4 clearly localizes to a specific subset of presynaptic neurotransmitter release sites in the *Drosophila* brain, and it is essential for the proper organization of presynaptic machinery at those sites. They show that loss of LRP4 reduces synapse number, whereas overexpression increases synapse number. Finally, they go on to show that LRP4 functions via the serine-arginine protein kinase SRPK79D, and LRP4 is required to maintain the proper localization of SRPK79D at presynaptic sites. The choice of experimental techniques consistently aims for the highest technical standard, and the data are of high technical quality. This study sheds new light on the mysterious CNS roles of a well-known NMJ synaptic organizer, and it sets a high standard for future work using *Drosophila* as a model organism in this field.

1) I do not think it is valid to conclude that "LRP4 preferentially localizes to *excitatory* neuron terminals", although I suppose this depends on the definition of "preferentially". Figure 1 shows that 59% of LRP+ somata in the vicinity of the antennal lobe are cholinergic, while 22% are glutamatergic, and essentially 0 are GABAergic. But there is solid evidence that glutamate inhibits antennal lobe neurons, and when inhibitory glutatamate-gated chloride channels are removed from these neurons, there is no physiological evidence of any residual excitation evoked by glutamate (Liu et al., 2013, Figure 3). Some of the somata in the vicinity of the antennal lobe might arborize in other neuropils (outside the antennal lobe) where glutamate has an excitatory role, but that is pure speculation at this point. We just don't know whether the LRP+ glutamatergic neurons are excitatory or inhibitory. Therefore, I would be hesitant to conclude that LRP preferentially localizes to excitatory terminals. Instead, it seems more accurate to say that LRP is expressed by a subset of cholinergic (excitatory) neurons and also a subset of glutamatergic neurons that might be excitatory or inhibitory. That said, the expression pattern of LRP is strikingly selective: it is excluded from GABAergic neurons.

2) The phenotypes associated with the *lrp4^dale^k* mutation are attributed to the authors to LRP's presynaptic roles: e.g., the authors write that "presynaptic LRP4 loss reduces synapse number as assayed both pre- and postsynaptically". Clearly, LRP does have presynaptic roles, given the antennal lobe phenotypes associated with ORN-specific perturbation, and also the lateral horn phenotypes associated with PN-specific perturbation. However, I don't see a good reason to exclude a postsynaptic role for LRP4. LRP4 is expressed by PNs, and we might expect it to traffic to PN dendrites, given that PN dendrites release neurotransmitter (Ng et al., 2002) and contain T-bars (Rybak et al., 2016; Tobin et al., 2017). It is possible that LRP4 is excluded from PN dendrites, and it functions in PN axons only. Alternatively, it is possible that LRP4 has a role in organizing neurotransmitter release sites in PN dendrites, but it has no role as a postsynaptic organizer in PN dendrites. Finally, the third alternative is that LRP4 plays a role in organizing postsynaptic sites in PN dendrites, potentially in addition to organizing presynaptic sites in presynaptic dendrites. A postsynaptic role for LRP4 in the Drosophlia brain is not implausible, given its well-documented role in organizing postsynaptic sites in mammalian muscle. I do not think the authors need to perform new experiments to address this issue. However, if they already have data on hand which could discriminate between these three alternatives, they should show it. And if they don't, then they should explicitly discuss the possibility that the phenotypes associated with the *lrp4^dalek^* mutation could reflect roles for postsynaptic LRP4 as well as presynaptic LRP4.

*Reviewer #2:*

The authors study LRP4, previously well studied in post-synaptic systems in the neuromuscular junction in mice. Here, they show that LRP4 has important presynaptic functions in the fly brain. They start by showing that LRP4 localises to neuron terminals at or near active zones. They say that this localization is preferential to excitatory terminals. It is striking that there is no localization to some inhibitory terminals. But, whether the localization seen is specific to excitatory terminals only is not fully proven by experiments. The authors may want to temper this conclusion.

The authors then examine the loss of function of presynaptic LRP4. This results in the reduction of synapse number, specific to excitatory synapses. When they examine olfactory behaviour after loss of function in ORN they remarkably find that "A complete loss of olfactory attraction was unexpected for a manipulation that reduced synapse number by ~30%." This is an interesting and unexpected result and they hypothesise that inhibition of inhibitory receptors should rescue this phenotype and this is tested elegantly. Gain-of function experiments have the expected 'opposite effect', but in both excitatory and inhibitory neurons. Learning from the fly NMJ the and the "similarity in the transverse nerve phenotypes and the role of SRPK79D at peripheral synapses, we hypothesised that LRP4 and SRPK79D could operate together in the CNS to regulate synapse number." They test the role of SRPK79D and find that it requires LRP4 and show that they interact proximity ligation assays.

In summary, this is an important study, very thorough, and whose results are a significant scientific advance for a general audience.

---

## [Author Response]

*Reviewer #1:*

*[…] 1) I do not think it is valid to conclude that "LRP4 preferentially localizes to excitatory neuron terminals", although I suppose this depends on the definition of "preferentially". Figure 1 shows that 59% of LRP+ somata in the vicinity of the antennal lobe are cholinergic, while 22% are glutamatergic, and essentially 0 are GABAergic. But there is solid evidence that glutamate inhibits antennal lobe neurons, and when inhibitory glutatamate-gated chloride channels are removed from these neurons, there is no physiological evidence of any residual excitation evoked by glutamate (Liu et al., 2013, Figure 3). Some of the somata in the vicinity of the antennal lobe might arborize in other neuropils (outside the antennal lobe) where glutamate has an excitatory role, but that is pure speculation at this point. We just don't know whether the LRP+ glutamatergic neurons are excitatory or inhibitory. Therefore, I would be hesitant to conclude that LRP preferentially localizes to excitatory terminals. Instead, it seems more accurate to say that LRP is expressed by a subset of cholinergic (excitatory) neurons and also a subset of glutamatergic neurons that might be excitatory or inhibitory. That said, the expression pattern of LRP is strikingly selective: it is excluded from GABAergic neurons.*

We agree with the reviewers’ caveats to our statement regarding the preferential localization of LRP4 to excitatory terminals. Therefore, we have adjusted the language in appropriate sections to highlight three major points: 1) the striking exclusion of LRP4 from GABAergic neurons, 2) the cholinergic excitatory identity of a majority of LRP4+ cells, and 3) the fact that our data cannot completely rule out that the subset of LRP4+ neurons are inhibitory. See Abstract; subsection “LRP4 is a synaptic protein expressed in excitatory neurons”, last paragraph; Discussion, first paragraph and subsection “LRP4 as a synaptic regulator that distinguishes excitatory from inhibitory presynaptic terminals”, last paragraph, for specific changes.

*2) The phenotypes associated with the lrp4^dale^k mutation are attributed to the authors to LRP's presynaptic roles: e.g., the authors write that "presynaptic LRP4 loss reduces synapse number as assayed both pre- and postsynaptically". Clearly, LRP does have presynaptic roles, given the antennal lobe phenotypes associated with ORN-specific perturbation, and also the lateral horn phenotypes associated with PN-specific perturbation. However, I don't see a good reason to exclude a postsynaptic role for LRP4. LRP4 is expressed by PNs, and we might expect it to traffic to PN dendrites, given that PN dendrites release neurotransmitter (Ng et al., 2002) and contain T-bars (Rybak et al., 2016; Tobin et al., 2017). It is possible that LRP4 is excluded from PN dendrites, and it functions in PN axons only. Alternatively, it is possible that LRP4 has a role in organizing neurotransmitter release sites in PN dendrites, but it has no role as a postsynaptic organizer in PN dendrites. Finally, the third alternative is that LRP4 plays a role in organizing postsynaptic sites in PN dendrites, potentially in addition to organizing presynaptic sites in presynaptic dendrites. A postsynaptic role for LRP4 in the Drosophlia brain is not implausible, given its well-documented role in organizing postsynaptic sites in mammalian muscle. I do not think the authors need to perform new experiments to address this issue. However, if they already have data on hand which could discriminate between these three alternatives, they should show it. And if they don't, then they should explicitly discuss the possibility that the phenotypes associated with the lrp4^dalek^ mutation could reflect roles for postsynaptic LRP4 as well as presynaptic LRP4.*

Indeed, this is correct. We cannot rule out a postsynaptic role for LRP4, nor is it our intent to do so! Rather, we focus on the presynaptic role for this organizer. It is unlikely that LRP4 is excluded from PN dendrites as there is robust LRP4-HA localization in the region of PN dendrites when expressed via *lrp4-GAL4* (Figure 1—figure supplement 2). In fact, we hypothesize that the second alternative posted by the reviewer is correct: that any LRP4 localizing to PN dendrites may serve a role as a presynaptic organizer for dendrodendritic synapses that reside there. Finally, the possibility of LRP4 as a postsynaptic protein remains present. Though we did not observe phenotypes consistent with that interpretation with PN-specific RNAi, it is difficult to conclusively interpret negative data. To acknowledge and discuss these alternative interpretations, we have adjusted the Results and Discussion sections where appropriate. See subsection “Perturbing presynaptic LRP4 changes ORN synapse number”, bottom of third paragraph and subsection “LRP4 function across evolution”, second paragraph, for specific changes.

*Reviewer #2:*

*The authors study LRP4, previously well studied in post-synaptic systems in the neuromuscular junction in mice. Here, they show that LRP4 has important presynaptic functions in the fly brain. They start by showing that LRP4 localises to neuron terminals at or near active zones. They say that this localization is preferential to excitatory terminals. It is striking that there is no localization to some inhibitory terminals. But, whether the localization seen is specific to excitatory terminals only is not fully proven by experiments. The authors may want to temper this conclusion.*

The reviewer raises a valid point; we have altered the language of our assertions (as above) to highlight alternative possibilities. As for reviewer 1, please see Abstract), subsection “LRP4 is a synaptic protein expressed in excitatory neurons”, last paragraph, Discussion, first paragraph) and subsection “LRP4 as a synaptic regulator that distinguishes excitatory from inhibitory presynaptic terminals”, last paragraph, for specific changes.